



# Environmental controls of non-growing season carbon dioxide fluxes in boreal and tundra environments

Alex Mavrovic[1-2-3-4], Oliver Sonnentag[2-4], Juha Lemmetyinen[5], Carolina Voigt[4-6], Nick Rutter[7], Paul Mann[7], Jean-Daniel Sylvain[8], Alexandre Roy[1-2]

[1] Université du Québec à Trois-Rivières, Trois-Rivières, Québec, G9A 5H7, Canada
  [2] Centre d'Études Nordiques, Québec, Québec, G1V 0A6, Canada
  [3] Polar Knowledge Canada, Canadian High Arctic Research Station campus, Cambridge Bay, Nunavut, X0B 0C0, Canada
  [4] Université de Montréal, Montréal, Québec, H3T 1J4, Canada
[5] Finnish Meteorological Institute, Helsinki, FI-00560, Finland
  [6] University of Eastern Finland, Kuopio, 70211, Finland
  [7] Northumbria University, Newcastle upon Tyne, NE1 8ST, UK
  [8] Ministère des Ressources naturelles et des Forêts, Québec, Québec, G1H 6R1, Canada

*Correspondence to:* Alex Mavrovic (Alex.Mavrovic@uqtr.ca)

**Abstract.** The carbon cycle in Arctic-boreal regions (ABR) is an important component of the planetary carbon balance, with growing concerns about the consequences of ABR warming on the global climate system. The greatest uncertainty in annual carbon dioxide ($CO_2$) budgets exists during the non-growing season, primarily due to challenges with data availability and limited spatial coverage in measurements. The

goal of this study was to determine the main environmental controls of non-growing season $CO_2$ fluxes in ABR over a latitudinal gradient (45ºN to 69ºN) featuring four different ecosystem types: closed-crown coniferous boreal forest, open-crown coniferous boreal forest, erect-shrub tundra, and prostrate-shrub tundra. $CO_2$ fluxes calculated using a snowpack diffusion gradient method ($n = 560$) ranged from 0 to 1.05 gC m$^2$ day$^{-1}$. To assess the dominant environmental controls governing $CO_2$ fluxes, a Random Forest machine

learning approach was used. We identified that soil temperature as the main control of non-growing season $CO_2$ fluxes with 68% of relative model importance, except when soil liquid water occurred during zero-degree Celsius curtain conditions ($T_{soil} \approx 0°C$ and liquid water coexists with ice in soil pores). Under zero-curtain conditions, liquid water content became the main control of $CO_2$ fluxes with 87% of relative model importance. We observed exponential regressions between $CO_2$ fluxes and soil temperature (RMSE = 0.024

gC m$^{-2}$ day$^{-1}$) in frozen soils, as well as liquid water content (RMSE = 0.137 gC m$^{-2}$ day$^{-1}$) in zero-curtain conditions. This study is showing the role of several variables on the spatio-temporal variability of $CO_2$ fluxes in ABR during the non-growing season and highlight that the complex vegetation-snow-soil interactions in northern environments must be considered when studying what drives the spatial variability of soil carbon emission during the non-growing season.


**Keywords:** Arctic-boreal regions, Carbon balance, Carbone dioxide flux, Random forest, Non-growing season, Soil temperature, Soil liquid water content, Snow.



# 1 Introduction

Carbon stocks and fluxes in the Arctic and boreal biomes (hereafter called Arctic-boreal regions; ABR) constitute large components of the planetary carbon balance (Tarnocai et al., 2009; van Huissteden and Dolman, 2012; Carreiras et al., 2017). ABR store substantial quantities of carbon due to inherently slow decomposition rates, largely attributable to cold temperatures (Ravn et al., 2020). ABR are warming up to four times faster than the rest of the planet with potential feedbacks to the global climate system (Derksen et al., 2019; Rantanen et al., 2022). Although ongoing warming of ABR has the potential to lengthen growing seasons, enhance plant growth and increase above-ground carbon storage (Sturm et al., 2005; McMahon et al., 2010), the growing season vegetation response is variable and complex (Myers-Smith et al., 2020). Warmer air and soil temperatures enhance production and release of carbon dioxide ($CO_2$) from ecosystem respiration, comprising heterotrophic respiration by microbes decomposing soil organic matter, and autotrophic respiration by above- and belowground plant components (Bond-Lamberty and Thomson, 2010). The release of previously frozen carbon stocks is particularly important in regions undergoing permafrost thaw (ground completely frozen for at least two consecutive years) (Schuur et al., 2015; Natali et al., 2021; Miner et al., 2022). If increases in ecosystem respiration exceed those of photosynthetic $CO_2$ uptake from enhanced plant growth, ABR may shift from a weak net $CO_2$ sink to a net $CO_2$ source, thereby generating a potentially non-negligible, positive feedback to the global climate system (Hayes et al. 2011; Gauthier et al., 2015; Natali et al., 2019; Bruhwiler et al., 2021; Virkkala et al., 2021; Braghiere et al., accepted).

During winter months in ABR, landscapes are generally snow-covered, photosynthesis is considered negligible, and therefore non-growing season $CO_2$ fluxes derive primarily from soil respiration (Christiansen et al., 2012; Webb et al., 2016). It is expected that complex soil-vegetation-snow interactions will lead to regional and local variability in soil respiration rates across ABR because of relationships between vegetation types, snow cover, soil properties, soil moisture and soil temperature (Gouttevin et al., 2012; Busseau et al., 2017; Loranty et al., 2018; Grünberg et al., 2020; Royer et al., 2021). Higher soil temperatures promote microbial activity and increase $CO_2$ production from soil organic matter decomposition during the non-growing season (Natali et al., 2019). A snowpack acts as an important thermal insulative layer for the soil during winter, keeping soils warmer than the ambient air (Dominé et al., 2016a). Vegetation affects snow properties by increasing snow depth where wind trapping occurs (Callaghan et al., 2011a; 2011b; Busseau et al., 2017), decreasing snow density and thermal conductivity around shrubs (Gouttevin et al., 2012; Dominé et al., 2015; 2016b), decreasing albedo due to protruding branches (Ménard et al., 2012), and causing earlier spring snowmelt due to vegetation thermal conductivity (Wilcox et al., 2019; Kropp et al., 2022). However, Dominé et al. (2022) showed that shrub branches within the snowpack can contribute to mid-winter soil cooling by conducting temperature through the snowpack. Hence, the complex vegetation-snow-soil interactions in northern environments must be considered when studying what drives the spatial variability of soil carbon emission during the non-growing season. Soil microbial activity can also be limited by lack or saturation of available water, meaning that higher amounts of available soil liquid water (LWC) should allow



higher heterotrophic respiration rates by increasing soil microbial activity as long as the soil environment is not anaerobic (Linn and Doran, 1984; Knowles et al., 2015). Anaerobic soil conditions are usually found in fully water saturated soils.

High uncertainties in non-growing season ABR $CO_2$ exchange between the ground surface and
atmosphere are in part due to limited data availability because of difficulties in accessing these vast, remote regions and the harsh winter conditions creating technical challenges for $CO_2$ fluxes measurements (Natali et al., 2019; Virkkala et al., 2022). Methods currently available to measure wintertime $CO_2$ fluxes include: 1) the eddy covariance technique (Baldocchi et al., 2003), 2) chamber measurements under or above the snowpack (McDowell et al., 2000) and 3) snowpack gradient diffusion methods (Sommerfield et al., 1993).
Each of these has their advantages and limitations. The eddy covariance technique (EC) exploits the atmosphere's turbulent nature to estimate net $CO_2$ fluxes at high temporal resolution without environmental disturbance (Baldocchi et al., 2001; Pastorello et al., 2020). Data gaps are common during the ABR winter since the EC equipment is energy-intensive and prone to failure in low temperatures. In addition, solar power supply systems are limited by low sunlight (Jentzsch et al., 2021, Pallandt et al., 2022). Furthermore, the EC
equipment is stationary and cover a large footprint (250-3000 m). In contrast, plot-scale chamber techniques for measuring $CO_2$ fluxes are portable methods with a small footprint (< 1m) (Subke et al., 2021; Maier et al., 2022). Chambers can be used either above the snowpack or directly on the ground. Placing a chamber on the snowpack does not provide a direct measurement of soil $CO_2$ fluxes due to $CO_2$ retention and lateral diffusion within snowpacks, generally creating a negative bias and uncertainties linking the snow/atmosphere
fluxes to soil fluxes (McDowell et al., 2000; Björkman et al., 2010a; Webb et al., 2016). Chambers can also be placed directly on the ground by excavating the snow cover (Elberling et al., 2007), providing a direct measurement of soil $CO_2$ fluxes that is, however, prone to a positive bias generated by a tunnel effect due to the snow excavation (McDowell et al., 2000; Björkman et al., 2010). Unavoidable snow cover disturbance also reduces the possibility of revisiting locations for temporal surveys because the soil thermal regime is
altered by the snow disturbance. Alternatively, permanent chambers can be installed before the first snowfall, but it disturbs the state of the ground and snow cover around the chamber (Webb et al., 2016). The snowpack diffusion gradient method uses snow porosity and tortuosity to estimate $CO_2$ fluxes from the gas concentration gradient along a vertical snow profile including ambient air above the snowpack (Sommerfield et al., 1993; Pirk et al., 2016; Kim et al., 2019). In this study, the snowpack diffusion gradient method will
be used to evaluate the spatial variability of $CO_2$ fluxes in ABR because of its portability and minimal environmental disturbance.

The goal of this study was to determine the main environmental controls of non-growing season $CO_2$ fluxes in ABR. 560 snowpack diffusion gradient measurements were made over a latitudinal gradient of four
different ecosystem types common in ABR in Canada: closed-crown coniferous boreal forest, open-crown coniferous boreal forest, erect-shrub tundra and prostrate-shrub tundra. Spatio-temporal measurements of snowpack $CO_2$ diffusion gradients were performed at several locations in four sites during the 2020-2021



and 2021-2022 winters (December to May). Firstly, a Random Forest (RF) machine learning analysis was used to evaluate the relative importance of the following environmental variables known to exert control over non-growing season $CO_2$ fluxes: soil temperature, soil LWC, vegetation type, snow water equivalent, snow depth and several snow density-related measurements. Secondly, the response and uncertainty of non-growing season $CO_2$ fluxes to the most impactful environment variables determined by the RF model were quantified through regression analysis.

## 2 Method

### 2.1 Study sites

To cover different vegetation types and a wide range of soil temperature ($T_{soil}$) regimes and snow conditions found in ABR, four study sites were selected across Canada (Fig. 1; Table 1). Each site represents a specific ecosystem type (Royer et al., 2021), and vegetation types within each of those ecosystems were determined using vegetation maps specific to each site. Cambridge Bay (CB), situated on the Victoria Island in the Canadian Archipelago was the northernmost site located in the Arctic tundra dominated by lichen and prostrate-shrub tundra. Ponomarenko et al. (2019) generated a detailed ecotype map of the Arctic tundra biome present in the CB study area. Here, these ecotypes were grouped by water availability into three tundra vegetation types from which the sampling locations (S) were selected: dry (S=94), sub-hydric (S=24) and hydric (S=110). Trail Valley Creek (TVC), NWT, situated just north of the treeline in the transitional zone between the boreal and Arctic biomes close to the Mackenzie delta, is dominated by erect-shrub tundra with remaining tree patches (Martin et al., 2022). Grünberg et al. (2020) produced a vegetation map of the TVC study area using airborne orthophotos, vegetation height and field observations from which seven vegetation types and landforms were identified: lichen (S=68), tussock (S=21), dwarf shrub (S=19), tall shrub (S=26), polygon (S=21), riparian shrub (S=17) and black spruce tree patch (S=18). Havikpak Creek (HPC) is located just south of the treeline, at about 50 km south of TVC in an open-crown black spruce dominated forest constituting the only type of vegetation present (S=30) (Krogh et al., 2017). Montmorency Forest (MM) is the southernmost site located in a closed-crown balsam fir dominated boreal forest constituting the only type of vegetation present (S=110) (Barry et al., 1988). The CB, TVC and HPC sites are underlain by continuous permafrost, while the MM site is permafrost-free.





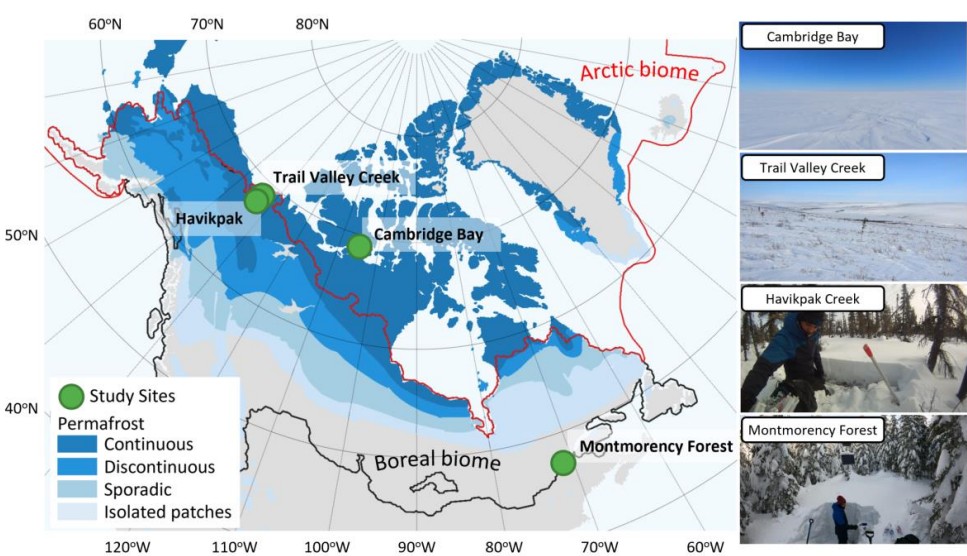

**Figure 1:** Study site locations in Canada. The Arctic biome is delimited following the Conservation of Arctic Flora and Fauna (CAFF) working group of the Arctic Council and the boreal biome is delimited following Potapov et al. (2008). Permafrost extent (Brown et al., 2002) is estimated in percent area: continuous (>90-100%), discontinuous (>50-90%), sporadic (10-50%) and isolated patches (<10%).

**Table 1:** Study sites with the number of sampling locations in Canada and $CO_2$ flux measurement (N) for each site.

| Site | Province/ Territory | Latitude/ longitude | Ecosystem | Sampling locations | N | Measurement Dates (YYYY-MM) | Site reference |
|---|---|---|---|---|---|---|---|
| Cambridge Bay | Nunavut | 69°13'N 104°54'W | Prostrate tundra shrubs | 47 | 230 | 2021-04, 12 2022-01 to 05 | Ponomarenko et al., 2019 |
| Trail Valley Creek | Northwest Territories | 68°46'N 133°28'W | Erect tundra shrubs | 34 | 190 | 2021-03, 12 2022-03 | Grünberg et al., 2020 |
| Havikpak Creek | Northwest Territories | 68°19'N 133°31'W | Open-crown coniferous boreal forest (black spruce) | 5 | 30 | 2021-03, 04 2022-03 | Krogh et al., 2017 |
| Montmorency Forest | Quebec | 47°18'N 71°10'W | Closed-crown coniferous boreal forest (balsam fir) | 12 | 110 | 2021-01, 02, 12 2022-01 to 05 | Barry et al., 1988 |

### 2.2 Snowpack diffusion gradient method

**2.2.1 $CO_2$ flux calculation**

During the non-growing season in ABR, soil respiration produces $CO_2$ below the snowpack. Consequently, a vertical $CO_2$ diffusion gradient is maintained through the snowpack ($d[CO_2]/dz$; gC m$^{-4}$) (Jones et al., 1999). The snowpack diffusion gradient method uses the $d[CO_2]/dz$ within the snowpack and Fick's first law for gas diffusion through porous media to estimate $CO_2$ fluxes ($F_{CO2}$; gC m$^{-2}$ day$^{-1}$) 155 (Sommerfeld et al., 1993; Zhu et al., 2014):





$$F_{CO2} = -\varphi\tau D \frac{d[CO_2]}{dz}$$ (1)

where $\varphi$ represents the porosity of the snow medium, $\tau$ its tortuosity and D the diffusion coefficient
of the diffused gas in $m^2$ $day^{-1}$. The porosity of dry snow can be assessed from its density (Kinar and Pomeroy,
2015):

$$\varphi = 1 - \frac{\rho_{snow}}{\rho_{ice}}$$ (2)

where $\rho$ represents the density of snow and pure ice ($\rho_{ice}$ = -0.0001·$T_{ice}$ + 0.9168 with $T_{ice}$ as ice
temperature in °C and $\rho_{ice}$ in g $cm^{-3}$; Harvey et al., 2017). The tortuosity is strongly correlated with porosity.
Prieur du Plessis and Masliyah (1991) established the following relationship:

$$\tau = \frac{1-(1-\varphi)^{2/3}}{\varphi}$$ (3)

Tortuosity can also be approximated as $\tau \approx \varphi^{1/3}$ (Millington, 1959; Mast et al., 1998). The
$d[CO_2]/dz$ is obtained by measuring the $[CO_2]$ vertical profile at various snow depths. Standard diffusion
coefficients of $CO_2$ are available in literature but must be corrected for temperature and pressure (Marrero
and Mason, 1972; Massman, 1988):

$$D = 0.2020 \cdot \left(\frac{T}{T_o}\right)^{1.590} \cdot e^{-\frac{0.3738}{T/T_o}}$$ (4)

where T is the air temperature and $T_o$ is the freezing point in K. The diffusion gradient method
assumes that gas fluxes are the result of simple, linear, gradient-induced diffusion in uniform porosity through
snow cover (McDowell et al., 2000). A snowpack with strongly heterogeneous density (i.e., vertical
stratification) can induce a bias when gas flow is altered by dense layers or ice crusts, typically leading to
$F_{CO2}$ overestimation (Seok et al., 2009). Such layers were rarely found in our study sites. The diffusion
gradient assumption also does not hold when strong wind events occur, decreasing snowpack $CO_2$
concentration through wind-pumping and inducing a negative bias on $CO_2$ fluxes (Seok et al., 2009).
Consequently, $d[CO_2]/dz$ was not measured in days following a strong wind event.

**2.2.2 Data collection**

All data were collected during the 2020-21 and 2021-22 winters between December and May (Table 1).
The $CO_2$ concentration gradient was measured by collecting gas samples at various depths in the snowpack.



Each gradient profile consisted of five gas samples collected at: 1) 5 cm above the snowpack (ambient air),
2) 5 cm depth below the snowpack surface, 3) 1/3 of total snow depth, 4) 2/3 of total snow depth and 5)
soil/snow interface. Gas present in snow pores was collected with a hollow, thin stainless-steel rod (50-120
cm long, 4 mm outer diameter and 2 mm inner diameter) to minimize snow disturbance (Fig. 2a). Gas was
collected in a 60 mL syringe (Air-Tite Luer Lock, Virginia Beach, Virginia) connected to the rod via a three-
way valve. Gases were transferred into 12 mL hermetic glass vials (Labco Exetainer®, Labco Ltd., Lampeter,
UK), which were sent to the Université du Québec à Trois-Rivières laboratory to be measured with a gas
analyzer to obtain $CO_2$ concentrations. At each site, several sampling locations were selected to cover the
maximum range of vegetation types and snowpack characteristics, covering areas of 0.05-22.5 km². At each
sampling location, 2 to 4 replicate profiles were measured at 50 cm spacing to test the repeatability of the
sampling. A minimal spacing of 5 – 7.5 cm was required between sampling positions since it corresponds to
the radius of the 60 ml sampling volume of each gas sample, based on a snow density range of 100 - 650 kg
m⁻³.

For typical Arctic snowpacks, samples at 1/3 depth are usually in wind slab, the dense and cohesive
surface snow layer formed by strong Arctic winds. Samples at 2/3 depth are usually in depth hoar, the lower
snow layer with low density and cohesion formed by a strong temperature gradient driving vertical vapor
flux through the snowpack (Fig. 2b). Typically, boreal snowpacks are deeper than in Arctic tundra and display
a more continuous vertical stratification with increasing snow density at the bottom of the snowpack. In HPC,
snowpack depths were 40-80 cm in March, while snowpack depths at MM were 100-200 cm (Fig. A1). For
comparison, by March, snowpacks at CB were 10-75 cm deep and 15-150 cm at TVC.

**Figure 2:** (a) Gas sampling equipment for the $CO_2$ concentration gradient measurement. (b) Typical snow depth profile
of an Arctic snowpack (picture from Trail Valley Creek close to a tree patch).

Once the gas samples were collected, a vertical profile of snow and soil properties was measured to
calculate snow porosity, tortuosity and the $CO_2$ diffusion coefficient. Snow properties included snow and soil



temperature (Snowmetrics digital thermometer; Fort Collins, Colorado; tenth of a degree resolution), snow density (Snowmetrics digital scale, 100 and 250 cm$^3$ snow cutters; $\sigma(\rho_{snow}) \approx 9\%$; Proksch et al., 2016) and snow stratigraphy. $T_{soil}$ was measured at 1 cm depth under the soil/snow interface, three measurements of 220 $T_{soil}$ were averaged. Snow depth measurements were done with a ruler graduated every 1cm ($\sigma(d_{snow}) \approx$ 0.5cm).

The $CO_2$ concentration of 86% of gas samples were measured using a Licor LI-7810 $CH_4/CO_2/H_2O$ Trace Gas Analyzer (LI-COR Biosciences, Lincoln, Nebraska; $\sigma < 1\%$; N = 483). The gas samples were 225 passed through an open loop along a continuous flow of a 200 ppm $CO_2$ calibration gas (Linde Canada, Ottawa, Ontario, Canada). Based on a calibration curve using 0, 400 and 1000 ppm $CO_2$ calibration gases (Linde Canada), the $CO_2$ concentration of gas samples were calculated (detailed protocol: https://www.licor.com/documents/xst0ld9jozfby78bmpdqi9i7rmjjjjmg).

Randomly distributed gas samples collected during the 2020-21 winter were analyzed with a Picarro G2201-I CRDS gas analyzer (Picarro, Santa Clara, Californie; $\sigma < 0.1\%$; N = 26). $CO_2$ concentrations estimated from the LI-7810 and Picarro gas analyzers were not significantly different in their concentration range and distribution (Fig. A2; $R^2 = 0.92$). At TVC in March 2022, a portable LI-850 $CO_2/H_2O$ Gas Analyzer was used ($\sigma < 1.5\%$; N = 38), allowing for $CO_2$ concentrations to be measured on the same day as sample 235 collection (avoiding the need for bottling and transportation). $CO_2$ concentrations estimated from the LI-7810 and LI-850 gas analyzers were not significantly different in their concentration range and distribution (Fig. A2b; $R^2 = 0.82$).

### 2.2.3 Evaluation of $CO_2$ flux uncertainties

An uncertainty assessment was conducted to evaluate $CO_2$ flux precision based on the snowpack 240 diffusion gradient method. From sampling to flux estimation, several steps could add uncertainty to the results. Uncertainties can be subdivided into four sources: gas concentration estimates, gas transfer/transport/storage, evaluation of the snowpack $d[CO_2]/dz$ and snowpit measurements. Gas concentration uncertainties were evaluated from the gas analyzer precision as assessed by the manufacturer and testing using calibration gases. Six $CO_2$ reference gases of 400 ppm were bottled during two different 245 field campaigns and were processed among the gas samples from the snowpack to ensure the transfer, transport and storage protocol did not lead to sample contamination. The $d[CO_2]/dz$ uncertainties were evaluated with the standard deviation from the coefficient of determination ($\sigma = \sqrt{(1 - R^2)/(N - 1)}$; Bowley, 1928). Using $d[CO_2]/dz$ and snow density uncertainties, $F_{CO2}$ uncertainties were calculated with the min-max uncertainty propagation method.



### 2.3 Soil volumetric liquid water content at Montmorency Forest site

When the soil is under zero-degree Celsius curtain conditions, the soil temperature is around freezing point (0°C) and a mix of ice and liquid water coexists in the soil pore space because the phase transition between water and ice is slowed due to latent heat (Outcalt et al., 1990). Hence, liquid water content (LWC; m³/m³) and ice fractions can be used as a freezing/thawing indicator during the zero-curtain period. The MM study sites were equipped with TEROS 12 Soil Moisture Sensors (METER Group) at 5 cm depth. LWC was only monitored at the MM site since it was the only site where the $T_{soil}$ measurements indicates that the zero-curtain was maintained throughout the winter season, allowing liquid water in the soil throughout the winter. The Zhang et al. (2010) empirical soil liquid water and ice mixing model was used to calculate soil liquid water content ($m_{uw}$) and ice fraction from permittivity probes:

$$LWC = a \cdot \frac{\rho_b}{\rho_w} \cdot |T_{soil}|^{-b} \tag{5}$$

$$\ln a = 0.5519 \cdot \ln SSA + 0.2618 \quad ; \quad \ln b = -0.264 \cdot \ln SSA + 0.3711 \tag{6}$$

where $\rho_w$ and $\rho_b$ (g cm⁻³) represent liquid water and soil bulk density respectively, $T_{soil}$ (°C) represents soil temperature, SSA (m⁻¹) represents soil particles specific surface area described by Fooladman (2011).

$$SAA = 3.89 \cdot d_g^{-0.905} \tag{7}$$

$$\ln d_g = f_c \cdot \ln M_c + f_{si} \cdot \ln M_{si} + f_{sa} \cdot \ln M_{sa} \tag{8}$$

where $d_g$ represents the soil geometric mean particle-size diameter (mm), $f$ and M represent soil fraction and mean particle-size diameter of soil components (mm). Soil components are clay ($M_c = 0.001$ mm), silt ($M_{si} = 0.026$ mm) and sand ($M_{sa} = 1.025$ mm). Soil bulk density and gravimetry was evaluated using a soil sampling protocol similar to the National Forest inventory protocol (CFI, 2008). Undisturbed soil samples were collected in three homogenous horizons of a soil profile using 400 cm² cores. Volumetric soil samples were dried (103°C) and weighted to determine bulk density. Gravimetric samples were used to determine sand (%, 50-2000 μm), silt (%, 2-50 μm), clay (%, < 2 μm) and organic content (g/kg). The soil texture was determined by the hydrometer method (Bouyoucos, 1962), whereas the organic content was determined with a LECO organic analysis instrument (LECO corporation, Saint-Joseph, Michigan).

### 2.4 Random Forest algorithm

Random forest (RF) is an ensemble machine learning method based on a decision trees (Breiman, 2001). Each decision tree of our RF model (scikit-learn 1.2.1 library from python 3.10.3) is trained on a random subset of environmental variables drawn from the dataset input: $T_{soil}$, LWC, vegetation type, snow

water equivalent, snow depth, snow mean density, snow maximum density, snow porosity, snow tortuosity, wind slab thickness (if present) and wind slab fraction relative to total snow depth (if present). Each decision tree generates a $F_{CO_2}$ prediction, and the overall RF prediction is the average of all prediction trees. A strength of the RF algorithm is that it performs well even when input variables are correlated with each other (Liaw and Wiener, 2002; Strobl et al., 2008; Kibtia et al., 2020). Our RF model was composed of 500 fully decomposed decision trees. Our dataset was randomly divided into a training subset (75%) and a testing

subset (25%), preserving the relative distribution between vegetation types. Our RF model performance was assessed using the coefficient of determination ($R^2$), explained variance, and mean absolute error. We used our RF model to identify the relative importance of non-growing season $CO_2$ flux predictors. Relative importance of each environmental variable was computed with the permutation method, i.e., alternatively removing variables from the RF model and evaluating the performance decrease which was, measured via

the coefficient of determination.

### 3 Results

### 3.1 $CO_2$ flux uncertainties

Evaluation of $F_{CO_2}$ precision showed that two main sources of uncertainty are associated with snow density measurements, in agreement with Sommerfeld et al. (1996), and with $d[CO_2]/dz$ linear regression

(mean $R^2$ = 0.790 ($\sigma$ = 0.236) for FCO2 ≥ 0.01 gC m$^{-2}$ day$^{-1}$; N = 398) (Table A1). Snow density uncertainty ($\sigma(\rho_{snow}) \approx 9\%$) impacted snow porosity and tortuosity in Eq. 1. From the linear fit of Fig. 3, the average $F_{CO_2}$ uncertainty can be estimated at 19.4%, which provides sufficient accuracy to observe the impact of environmental variables on non-growing season $F_{CO_2}$.

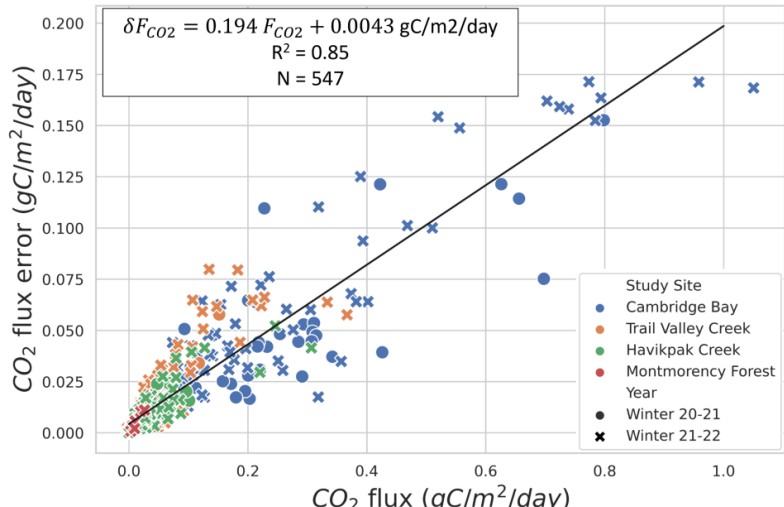

**Figure 3:** $CO_2$ flux ($F_{CO_2}$) uncertainty relationship to $F_{CO_2}$ for the four study sites and two winter seasons 2020-2021 and 2021-2022. Specifications of the linear fit can be found in the upper left.





The overall [CO$_2$] precision of around 1% shows that the measurement technique is not a main source of uncertainty in F$_{CO2}$ estimates. Gas concentration estimations from LI-7810 have a precision of 0.88% at 400 ppm according to the manufacturer. The stability of the [CO$_2$] measurement was evaluated over 169 measurements displaying a standard deviation of 0.09%. The LI-7810 was further tested using a 400 ppm calibration gas with a 1% [CO$_2$] precision (Linde Canada). A linear calibration fit equation was used to estimate [CO2] of small gas samples, using 3 calibration gases (200, 400, 1000 ppm) plus the theoretical zero intercept. Average uncertainty of the linear regression was 0.76% over six calibration runs with a standard deviation of 0.15%. The average accuracy of the reference [CO$_2$] bottled among the gas samples from the snowpack was 1.11%.

### 3.2 Spatio-temporal variability of non-growing season CO$_2$ fluxes associated with abiotic controls

The RF model determined T$_{soil}$ and LWC were the two main predictors of non-growing season CO$_2$ fluxes. We found two temperature and LWC regimes of non-growing season F$_{CO2}$ (Fig. 4). The first regime was when the soil was frozen with T$_{soil}$ < 0ºC and LWC < 0.2 m$^3$/m$^3$ leading to F$_{CO2}$ being mainly controlled by T$_{soil}$. The second regime was when LWC > 0.2 m$^3$/m$^3$ and < 0.42 m$^3$/m$^3$ but with a fraction of it's water in the form of ice (zero curtain condition), causing LWC to be the main control of F$_{CO2}$ instead of T$_{soil}$. Subsequent evaluation focused on the response of non-growing season CO$_2$ fluxes to T$_{soil}$ and LWC using exponential regressions in order to better understand the role of these two variables on non-growing season CO$_2$ fluxes.

### 3.2.1 Variable importance determined by Random Forest model

T$_{soil}$ was the F$_{CO2}$ predictor with the highest relative importance (68%) when using the complete dataset (Fig. 4a), followed by LWC (17%). Snowpack characteristics, $\rho_{snow}$ (11%) and SWE (2%), had a lower relative importance in the RF model. Contrary to what might be expected, the vegetation type had near-negligible relative importance (1%) in F$_{CO2}$ prediction. The RF model was developed starting with all environmental variables available: T$_{soil}$, LWC, vegetation type, snow water equivalent (SWE), snow depth, mean $\rho_{snow}$, max $\rho_{snow}$, $\varphi$, $\tau$, wind slab fraction and wind slab thickness. Although the correlation of several snow parameters did not decrease the RF model performance, snow parameters impacted the assessment of variable relative importance by splitting the relative importance between the correlated variables. Consequently, variables with lower importance and with no significant impacts on the RF performance were progressively removed. The two selected snow parameters that had significant impact were SWE and $\rho_{snow}$. MM was the only site where soil LWC was present, enabling the assessment of the relative importance of this variable. When using only data from MM in the RF model (Fig. 4b), the relative importance of T$_{soil}$ (12%) on F$_{CO2}$ was lower than with all combined datasets since T$_{soil}$ was near 0ºC for all measurements. At MM, LWC becomes the main predictor (87%) of F$_{CO2}$, while $\rho_{snow}$ importance drops (2%) and SWE remains similar (< 1%).



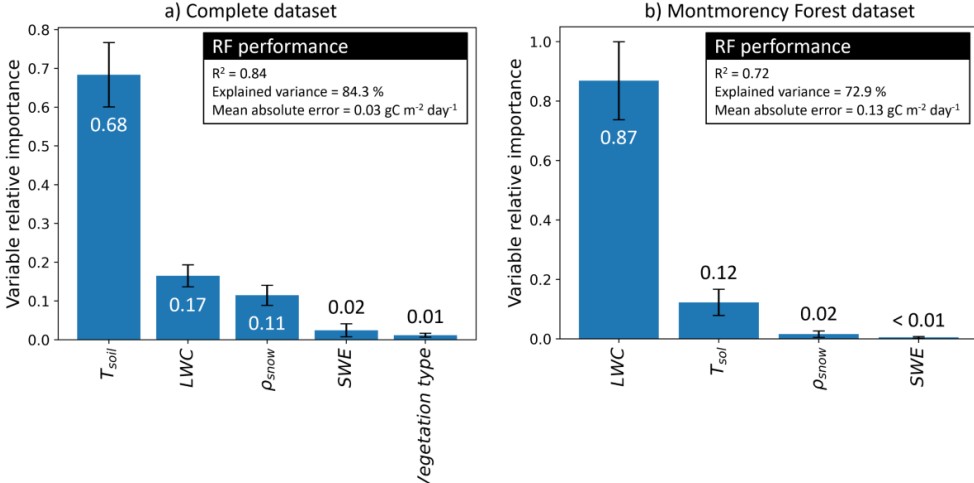

**Figure 4:** Random Forest (RF) performance and variable relative importance. Variables used are soil temperature ($T_{soil}$), soil liquid water content (LWC), snow density ($\rho_{snow}$), snow water equivalent (SWE) and vegetation type. (a) The first iteration integrated the complete dataset and (b) the second iteration only integrated Montmorency Forest dataset with LWC > 0 $m^3$ $m^{-3}$. The values displayed by the bar plot is the mean variable relative importance over 100 permutations, while the error bars are the standard deviation.

### 3.2.2 Soil temperature

Figures 5 and 6 show the relationship between non-growing season $F_{CO2}$ and $T_{soil}$. Figure 5 focuses on $T_{soil} < 0°C$ from CB, TVC, HPC and MM. An exponential regression was used to evaluate the relationship between $T_{soil}$ and $F_{CO2}$ estimates (RMSE = 0.024 gC $m^{-2}$ $day^{-1}$). $F_{CO2}$ at MM when $T_{soil} < 0°C$ and LWC < 0.2 $m^3/m^3$ were included in this graph because they are more strongly correlated to $T_{soil}$ than LWC (see Sect. 3.2.3). Note that the lack of $F_{CO2}$ measurements with $T_{soil}$ between -6°C to -0.5°C restrict the capacity to evaluate the regression within this range. Using the exponential regression of Natali et al. (2019), we obtained a RMSE of 0.030 gC $m^{-2}$ $day^{-1}$, higher than the one obtained with our dataset. The regression of Natali et al. (2019) generally shows an overestimation of fluxes for $T_{soil} < -5°C$, but an underestimation for $T_{soil} > 5°C$ when compared to our exponential regression. The systematic bias between our dataset and the regression of Natali et al. (2019) is minimal (mean bias = -0.0025 gC $day^{-1}$ $m^{-2}$). We also observed the isolated occurrence of comparably large, non-growing season $F_{CO2}$ up to 0.36 gC $m^{-2}$ $day^{-1}$ at temperatures below -10°C (Fig. 5). These measurements of high $F_{CO2}$ at low temperature seems to be genuine since the repeatability was verified over the 3 sampling profiles performed at each site. Nevertheless, we were not able to explain these strong $F_{CO2}$ fluxes and no environmental variables measured in our study could be linked to those occurrences. Figure 6 displays the higher non-growing season $F_{CO2}$ from MM where $T_{air}$ are higher and the important snowpack insulation keeps the soil at temperatures around 0°C through the entire non-growing season. $F_{CO2}$ increases rapidly with $T_{soil}$ above the freezing point, which is depicted in the exponential regression (RMSE = 0.286 gC $m^{-2}$ $day^{-1}$).

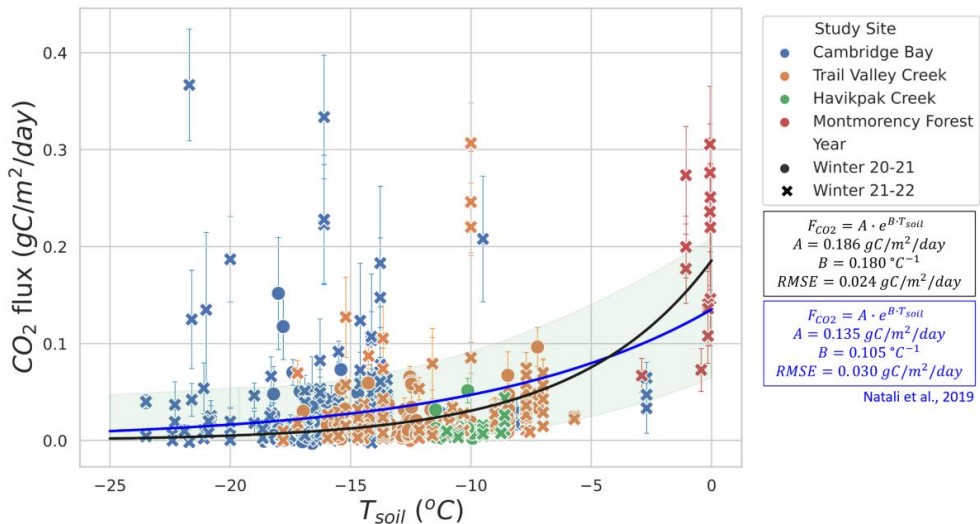

**Figure 5:** $CO_2$ flux ($F_{CO2}$) as a function of surface soil temperature ($T_{soil}$) for $T_{soil} < 0$ºC. An exponential regression was fitted with the data (black line) and compared to the exponential regression by Natali et al. 2019 from an external dataset (blue line).

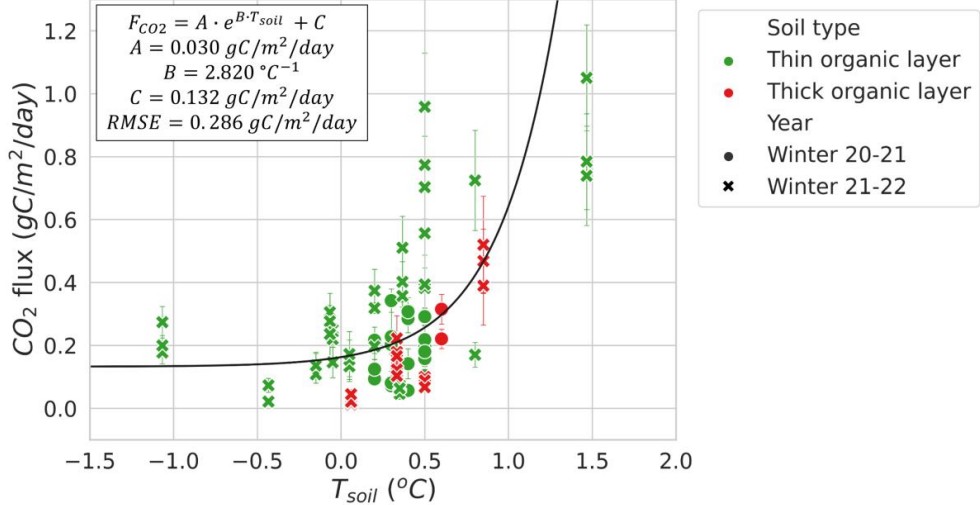

**Figure 6:** $CO_2$ flux ($F_{CO2}$) as a function of soil temperature ($T_{soil}$) at the Montmorency Forest study sites where soil liquid water content (LWC) was greater than 0 m³/m³ through the winter. An exponential regression was fitted to the data (black line).

### 3.2.3 Soil liquid water content

The relationship between LWC and $F_{CO2}$ during the non-growing season at MM (RMSE = 0.137 gC m⁻² day⁻¹) was stronger than between $T_{soil}$ and $F_{CO2}$ (RMSE = 0.286 gC m⁻² day⁻¹), when excluding the


sampling location that contained a thick organic soil layer with very high soil moisture due to its location near the bottom of a microtopographic depression (Fig. 7). Other MM sampling locations with a thin organic layer shared a similar soil compositions dominated by mineral soils. The strong correlation between LWC

and $F_{CO_2}$ was mostly observed at LWC $> 0.2$ m$^3$/m$^3$ and $< 0.42$ m$^3$/m$^3$. The plateau observed in Fig. 7 indicates that $T_{soil}$ might be a better predictor than LWC at LWC $< 0.2$ m$^3$/m$^3$.

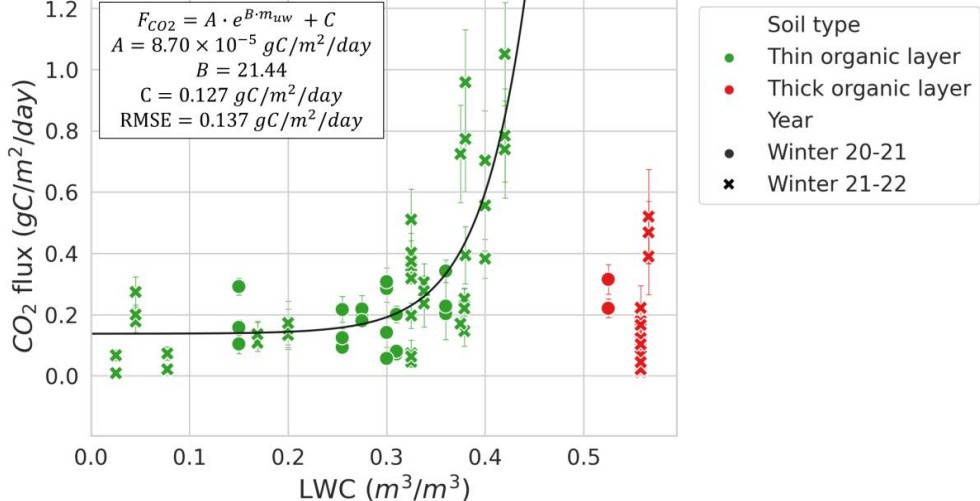

**Figure 7:** CO$_2$ flux ($F_{CO_2}$) as a function of soil volumetric liquid water content (LWC) at the Montmorency Forest study
sites. An exponential regression was fitted to the data (black line), excluding the thick organic layer site (red markers).

### 3.2.4 Vegetation types

Figure 8 shows non-growing season $F_{CO_2}$ across the four study sites for different vegetation types. Since CB vegetation is mostly prostrate-shrub tundra, CB ecosystems were regrouped by water availability.

Median non-growing season $F_{CO_2}$ at CB were observed in environments experiencing wetter conditions during the growing season. At TVC, several vegetation and land cover types are present. $F_{CO_2}$ from MM were higher than for the other sites. Higher $F_{CO_2}$ can be explained by warmer mean annual average temperature, a deeper snowpack and winter $T_{soil}$ around 0°C (See Sect. 3.4).

Vegetation type was not identified as a strong predictor of $F_{CO_2}$ by the RF model. Nonetheless, we observed differences in the mean and range of $F_{CO_2}$ for the various vegetation types probed in this study. This might be due to the strong correlation between vegetation type and soil temperature (Fig. A3), as well as relationships between vegetation and soil type, including soil organic matter content and soil pore size. The RF algorithm showed vegetation type relative importance increased to 42% when $T_{soil}$ was removed from the

environmental variables, although the removal of $T_{soil}$ decreased RF performance substantially ($R^2 = 0.40$).





Therefore, vegetation could be used as a proxy variable for $T_{soil}$ if the latter is not available to predict $F_{CO2}$, but with poorer results.

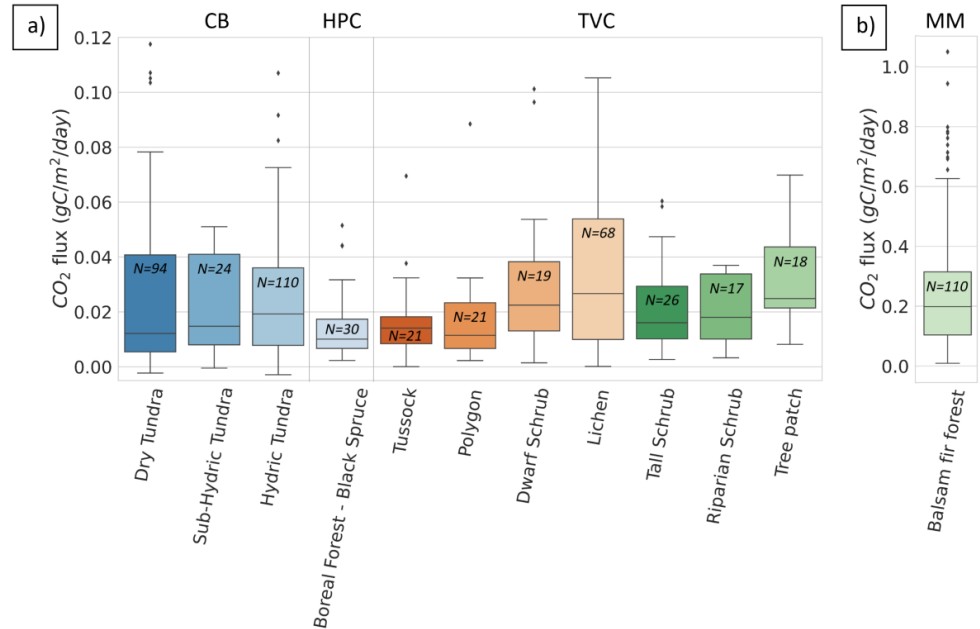


**Figure 8:** Boxplot of $CO_2$ flux ($F_{CO2}$) across 12 vegetation types and 4 sites. $F_{CO2}$ are on a separate scale (b) because they are much higher than the colder environments. Cambridge Bay (CB) sites are ordered by increasing water availability and Trail Valley Creek (TVC) sites are ordered by increasing soil surface temperature in March 2021 and 2022. Havikpak Creek (HPC) and Montmorency Forest (MM) were composed of a single vegetation type. Outliers were defined as $F_{CO2}$

> Q3 + 1.5 IQR where Q3 is the third quartile and IQR the interquartile range. Outliers are out of y-axis range for the dry tundra (4), sub-hydric tundra (4), hydric tundra (4) and lichen (3). The outliers can be found in Fig. 5.

## 4 Discussion

### 4.1 Controls of non-growing season $CO_2$ fluxes

The RF model supported that $T_{soil}$ was the dominant predictor of non-growing season $F_{CO2}$ when the soil

was frozen. However, in the closed-boreal forest site where the zero-curtain conditions remained throughout the winter, LWC became the most important predictor. Our results corroborate the strong non-growing season $F_{CO2}$ dependency on $T_{soil}$ shown by Natali et al. (2019), although we observed fluxes lower than reported by Natali et al. (2019) at $T_{soil} < -5$°C and mostly higher fluxes at $T_{soil} > -5$°C. It should be noted that the Natali et al. (2019) regression was obtained using $F_{CO2}$ estimates from several methods including eddy covariance,

chamber and snowpack diffusion measurements, whereas our study exclusively uses the latter. Several studies have shown bias between the different measurement methods; eddy covariance and soil chamber methods displayed positive biases when compared to snowpack diffusion measurements (McDowell et al.,





2000; Björkman et al., 2010a; Webb et al., 2016), while the snow chamber displayed negative biases when compared to the snowpack diffusion measurements (McDowell et al., 2000).

**4.2 Zero-curtain conditions**

Soil LWC was observed only at the MM site, where $T_{soil}$ was around 0ºC throughout winter. In zero-curtain conditions, LWC was shown to become the dominant control of non-growing season $F_{CO2}$, while $T_{soil}$ importance diminished. Our study highlighted the important impact of LWC on $F_{CO2}$ around soil freezing point when there is a mixed state of ice and free water in soil. When the soil is at zero-curtain, the latent heat

governs the ice and liquid water ratio in the soil (Devoie et al., 2022). Hence, LWC and ice fractions can be used as a freezing/thawing indicator during the zero-curtain period and help better quantify the $F_{CO2}$ fluxes in boreal forest environment where zero-curtain conditions prevail (Prince et al., 2019). This result is particularly important in ABR since the duration and frequency of zero-curtain periods are expected to increase in a warming climate (Yi et al., 2015 and 2019, Tao et al., 2021). It should be noted that one of the

measurement locations at MM displayed low $F_{CO2}$ despite its LWC being the highest of all sites. The soil composition of this site consisted of a thick (> 30 cm) soil organic top layer, whereas all other measurements were done at sites with thinner (3-10 cm) organic layers on top of mineral soil. It is well known that anaerobic conditions created by high soil moisture (at least > 50%) constrain soil $CO_2$ respiration rates during the growing season because many microorganisms require oxygen for organic matter decomposition which they

lack if soil porosities are filled with water (Linn and Doran, 1984; Davidson and Janssens, 2006).

**4.3 Snowpack importance**

Our study shows that abiotic variables related to $T_{soil}$, LWC, and physical snowpack properties explain the majority of variance in winter $CO_2$ fluxes. It should be noted that we did not incorporate variables related to temporal dynamics such as the previous days' soil temperature and LWC, which have been shown by Harel

et al. (2023) to be of importance during the growing season. However, winter soil variables are not expected to be as dynamic as during the growing season because of the snowpack insulating properties. The RF model showed that SWE and mean snow density were the snow characteristics that provided the greatest improvement of the RF model, although to a lesser degree than $T_{soil}$ and LWC. The importance of snow characteristics in $F_{CO2}$ is linked with the strong correlation to $T_{soil}$ (Dominé et al., 2016a), although a snapshot

of snow conditions provides limited abilities to infer $T_{soil}$ as shown in Slater et al. (2017). Snow properties temporal information is required to predict the impact of snow insulation on $T_{soil}$, with the most important period being in the autumn freeze-up when air temperature decreases below the freezing point. Snow characteristics are closely linked to topography (Meloche et al., 2021), and thus soil wetness and soil carbon content (Gouttevin et al., 2012).

**4.4 Soil biogeochemistry**

The unexplained variance (16%) suggests that winter $CO_2$ fluxes might have been controlled by other environmental variables such as soil physical-chemical properties regulating soil biogeochemistry and soil redox conditions, which were not addressed in this study. $CO_2$ production is governed by the availability of labile C compounds regulating the decomposition of soil organic matter (Michaelson et al., 2005; Wang et

al., 2011), and the activity and composition of the soil microbial community (Monson et al., 2006). Soil type and structure, for example the thickness of the organic layer, soil pore size distribution, as well as soil pH may be further strong controls on $CO_2$ production (Steponavičienė et al., 2022; Yli-Halla et al., 2022). All these variables vary widely across the heterogeneous tundra terrain (Virtanen and Ek, 2014), where small-scale moisture, vegetation and soil conditions occur among hummock and inter-hummock depressions

(Wilcox et al. 2019). Further analysis is required to understand the role of physical-chemistry soil properties on $F_{CO2}$ during the non-growing season.

**4.5 Relevance for terrestrial biosphere models**

Large uncertainties remain in terrestrial biosphere models used to estimate $CO_2$ fluxes in the ABR (Fisher et al., 2014; Tei and Sugimoto, 2020; Birch et al., 2021; Virkkala et al., 2021), especially regarding the

respiratory release of $CO_2$ via soil respiration (the sum of heterotrophic respiration and belowground autotrophic respiration) during the non-growing season (Natali et al., 2019). The limited number of observational data available has restricted model improvements, testing, and evaluation (Virkkala et al., 2022). Modelling the ABR carbon cycle is critical for climate projections since a warmer climate should lead to higher $T_{soil}$, thus increasing ABR non-growing season $F_{CO2}$ (Mellander et al., 2007; Throop et al., 2012;

Wieder et al., 2019). Several terrestrial biosphere models are currently in use (Fisher et al., 2022), such as CLM (Community Land Model; Lawrence et al., 2019) and CLASSIC (Canadian Land Surface Scheme Including Biogeochemical Cycles; Melton et al., 2020; Seiler et al., 2021). The $F_{CO2}$ relationships to $T_{soil}$ and LWC observed in this study could be used to inform terrestrial biosphere models through the parametrization of non-growing season soil respiration sensitivity to soil temperature (e.g., Q10) and LWC in zero-curtain

conditions. Our study shows that permanent installation of the snow gradient method (Seok at al., 2009; Zhu et al., 2014; Graham and Risk, 2018) would be suitable to gather temporal non growing season $CO_2$ fluxes in ABR required to fully test terrestrial biospheres models.

**5 Conclusion**

Our study showed that $T_{soil}$ is the main control of non-growing season $F_{CO2}$ at $T_{soil} < 0°C$ in ABR.

The relative importance analysis of our RF model showed that $T_{soil}$ was the main predictor of $F_{CO2}$, followed by LWC. However, we found that at our site maintaining zero-curtain conditions throughout winter, LWC becomes the main control of non-growing season $F_{CO2}$. We observed non-negligible non-growing season $F_{CO2}$ that may partially offset growing season $CO_2$ uptake in ABR. Consequently, non-growing season $F_{CO2}$



must be properly estimated in terrestrial biosphere models and climate models. Additionally, future research
should focus on linking the effects of abiotic variables on $F_{CO_2}$ during the non-growing season, as we
determined here, with soil biogeochemistry, microbial functioning and vegetation.

## 6 Acknowledgements

This work was made possible thanks to the contributions of the Natural Sciences and Engineering Research
Council of Canada (NSERC), the Fonds de recherche du Québec – Nature et technologies (FRQNT) and
Polar Knowledge Canada (POLAR). A special thanks to everybody that contributed to data collection and
gas analyzing: Elise Imbeau (Viventem), Gabriel Ferland (Viventem), Aili Pedersen (POLAR), Gabriel
Hould Gosselin (Université de Montréal [UdeM] and Wilfrid Laurier University [WLU]), Emma Riley
(WLU), Rosy Tutton (WLU), Victoria Dutch (Northumbria University [NU]), Georgina Woolley (NU), Élise
Groulx (Université de Sherbrooke [UdeS]), Charlotte Crevier (UdeS), Érika Boisvert (UdeS), Alain Royer
(UdeS), Patrick Ménard (UdeS), Vincent Sasseville (UdeS), Célia Trunz (UdeS), Daniel Kramer (UdeS),
Estéban Hamel Jomphe (UQTR), Samuel Goulet (UQTR), Alex Gélinas (UQTR), David de Courville
(UQTR), Juliette Ortet (UQTR) and Chris Derksen (Environment and Climate Change Canada). We would
also like to thank Ian Hogg, Johann Wagner and Scott Johnson from POLAR for their logistical support.

## 7 Competing interests

The authors declare that they have no conflict of interest.

## 8 Data availability

Mavrovic, A., Sonnentag, O., Voigt, C., Roy, A. (2023). Non-growing season CO2 fluxes over arctic and
boreal environments. Borealis, V1, UNF:6:u35JsrWpGNJSM3chSR9Ssg== [fileUNF]. doi:
10.5683/SP3/R3KZEQ



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





**Appendix A**

**Table A1: Uncertainty sources on F$_{CO2}$ and their uncertainty. [CO$_2$] precision was evaluated at a concentration of 400 ppm.**

| F$_{CO2}$ uncertainty source | Uncertainty |
|---|---|
| [CO$_2$] estimate | |
| · LI-7810 precision | 3,52 ppm (0.88%) |
| · Measurement stability | 3.6 ppm (0.09 %; N=169) |
| · Reference gas | 4 ppm (1 %) |
| · Calibration fit | 3.04 ppm (0.76 %; N=6; σ = 0.15) |
| · Transfer, transport and storage test | 4.44 ppm (1.11 %; N=6) |
| Snow density (kg/m³) | 9 % |
| d[CO$_2$]/dz linear regression (gC/m⁴) | 19.4 % |



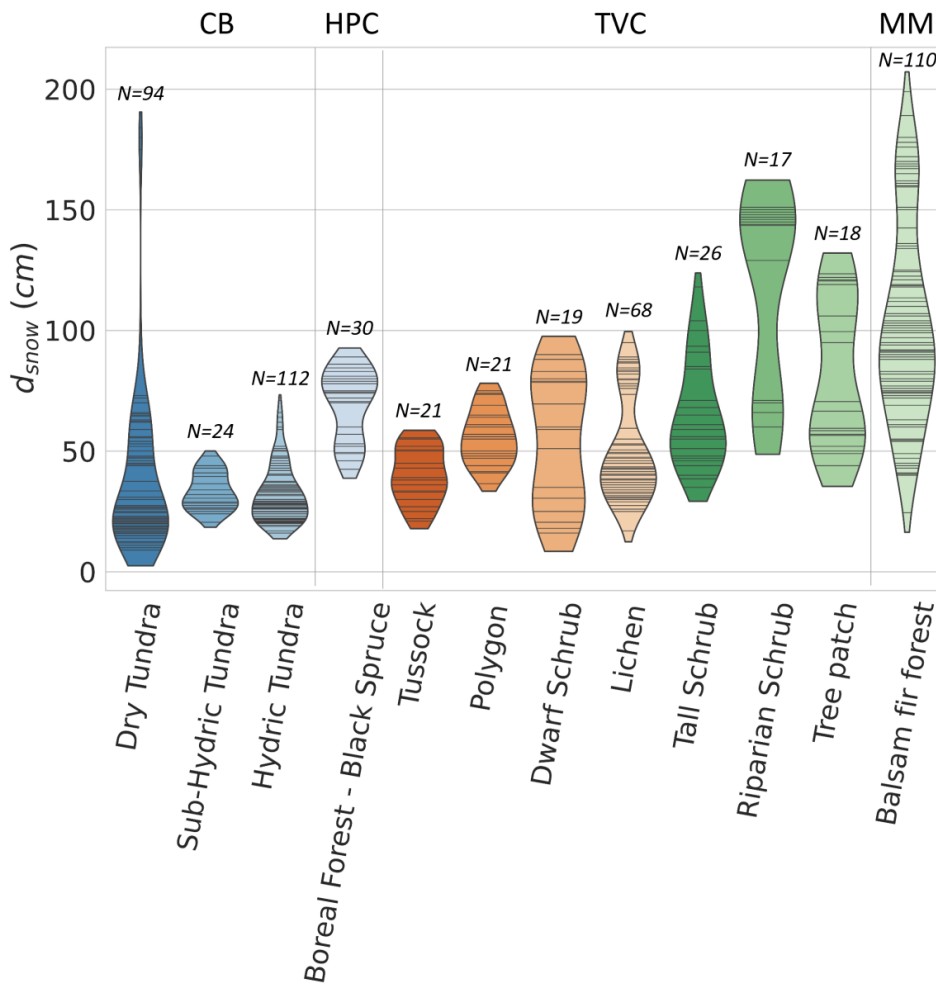

**Figure A1:** Violin plot of the snow depth range of sites where $F_{CO_2}$ was estimated. The black stripes inside the violins represent data point. CB sites are ordered by increasing hygricity and TVC sites are ordered by increasing soil surface temperature in March 2021 and 2022.



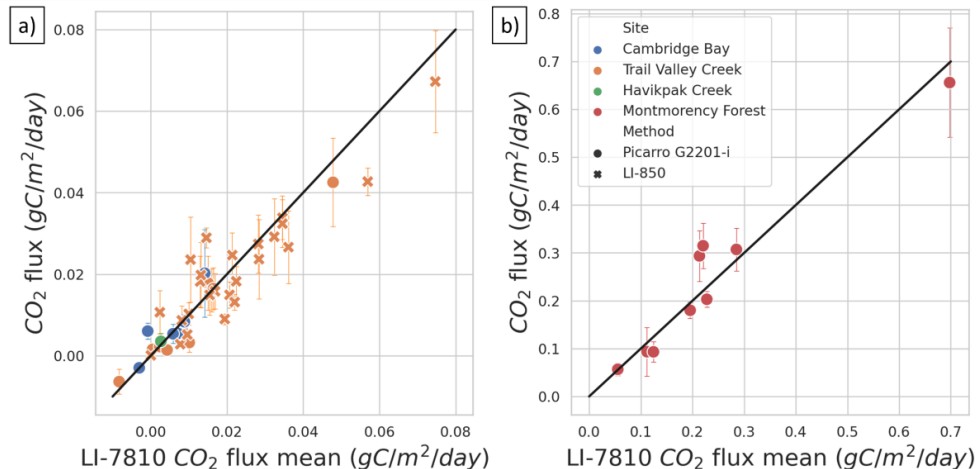

**Figure A2:** Comparison of non-growing season CO₂ flux calculated from CO2 concentration estimated with different gas analyzers. The LI-7810 gas analyzer was used as the reference and is compared to a Picarro G2201-*i* and LI-850. In the arctic biome (a), the correlation coefficient is 0.924 for the Picarro and 0.821 for the LI-850. In the boreal biome (b), the correlation coefficient is 0.929 for the Picarro.



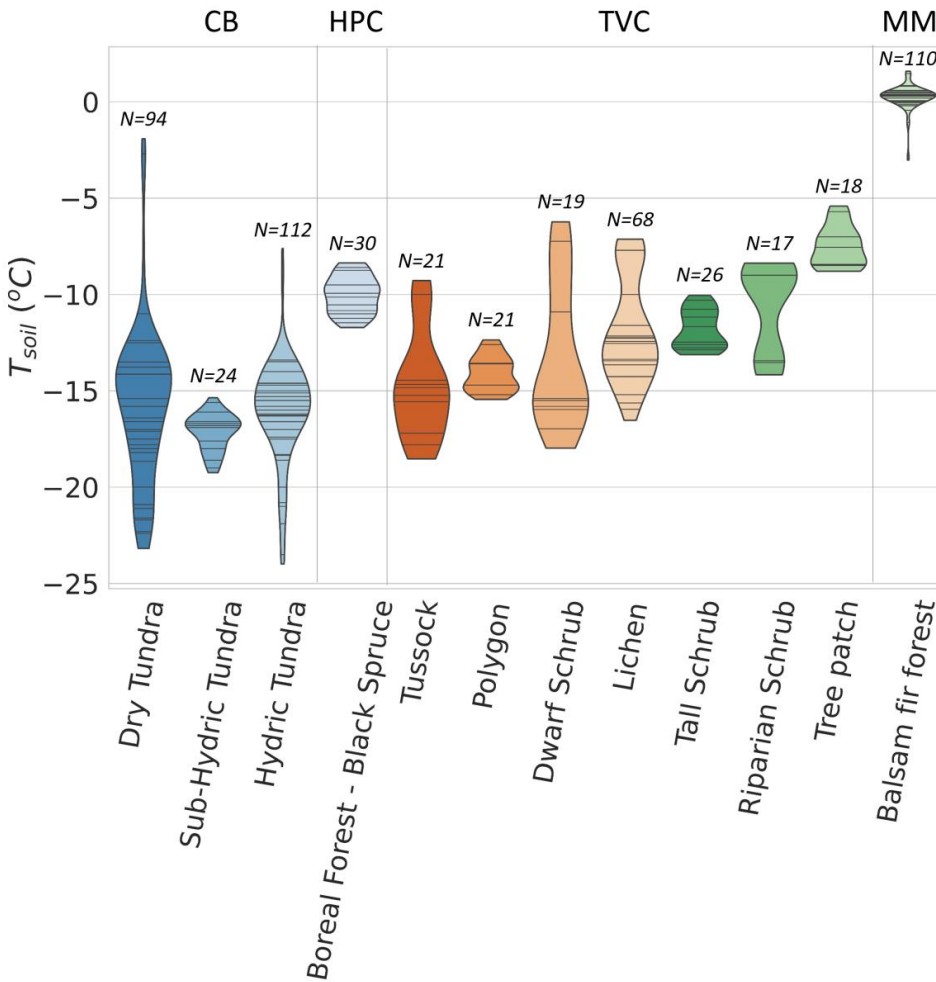

**Figure A3:** Violin plot of the soil temperature ($T_{soil}$) range of sites where $F_{CO2}$ was estimated. The black stripes inside the violins represent data point. CB sites are ordered by increasing hygricity and TVC sites are ordered by increasing soil surface temperature in March 2021 and 2022.