# Peer review of "Environmental controls of winter soil carbon dioxide fluxes in boreal and tundra environments"

_Biogeosciences, 2023_

## Author Comment (AC2)

In blue: Reviewer's comments. [ ] = Numbering

In black: Answers to referees.
P=Page; L=Line; Track change version
*In black and italic: Modification added to text.*

**Reviewer #2:**

Synopsis:

This manuscript reports data on non-growing season CO2 fluxes from 4 different sites in the Arctic-boreal region. Three sites are Arctic while one site is Boreal. Measurements of CO2 concentrations down through the snowpack were done over two consecutive years. Snow samples were used to infer the diffusion coefficient of the snow in order to calculate CO2 fluxes based on the concentration gradient using Fick's law of diffusion.

As the authors point out, there is still today a lack of data and understanding of what governs CO2 flux rates during the non-growing season in the Arctic. Based on the measurements performed the authors show that soil temperature is the dominant predictor of resulting CO2 fluxes at sub-zero temperatures across sites, whereas during zero-curtain conditions liquid water content becomes the primary predictor of CO2 flux. These two variables dominated also over e.g. vegetation type. This is an interesting result warranting publication. However, some issues should be resolved before the manuscript is ready for publication.

Thank you for this assessment of our study.

General comments:

[1] As such, the applied field methodology seems sound and well described except for a few critical details that should be explained further, including how many and in which depths snow samples were done for obtaining information about the snowpack conditions. The authors explain how the snow properties change down through the profile but it is unclear if this is reflected in the snow sampling procedure – e.g. how are changes down through the profile in terms of changes in diffusivity incorporated into the flux calculations.

The snowpack density was measured every 5 cm, it was specified in the manuscript. Although the density of the depth hoar and wind is typically different, the diffusion gradient was still linear. Therefore, the snowpack average density was used for the calculation.

P8, L222-229: Snow properties *were measured at every 5 cm including* snow temperature (Snowmetrics digital thermometer; Fort Collins, Colorado; tenth of a degree resolution), snow density (Snowmetrics digital scale, 100 and 250 cm$^3$ snow cutters *used to weigh snow samples*; $\sigma(\rho snow) \approx 9\%$; Proksch et al., 2016)*, snow liquid water content (hand*

*test from Fierz et al., 2009)* and snow stratigraphy. $T_{soil}$ was measured at 1 cm depth under the soil/snow interface *(Snowmetrics digital thermometer; Fort Collins, Colorado; tenth of a degree resolution)*, three measurements of $T_{soil}$ were averaged. Snow depth measurements were done with a ruler graduated every 1cm ($\sigma(d_{snow}) \approx 0.5$cm).

[2] More critically, it is clear that liquid water content was only measure at one site, but unclear if the liquid water content was then estimated at all the other sites except for MM, based on soil temperature and soil properties. It seems to me that this is what was done but it is unclear and this disturbs the overall understanding also of the RF modelling, where there may or may not be missing LWC estimates from 3 of the 4 sites. The authors should clarify explicitly which LWC data were available from all sites for the RF model. If data on LWC were indeed not estimated for the 3 other sites, and therefore missing in the RF modelling, the authors should explain how the RF model handles these missing values and how this affects the interpretation of the RF model result for the importance of LWC when all data are included in the analysis. Except for this lack of clarity, the analyses performed also seems sound.

LWC was only calculated at MM and was estimated as negligible (i.e., LWC ≈ 0) for all other sites. This assumption is supported by the model from Zhang et al. (2010) and served as input for the Random Forest model. This assumption was spelled out in the manuscript.

P9, L267-273: LWC was only monitored at the MM site since it was the only site where $T_{soil}$ *remained* around 0°C for the whole *non-growing season*, allowing *the presence of* liquid water in the soil throughout the *non-growing season*. The Zhang et al. (2010) empirical soil liquid water and ice mixing model was used to calculate soil liquid water content ($m_{uw}$) *(Eq. 5 to 8). LWC was estimated to be negligible at the CB, TVC and HPC sites since $T_{soil}$ was in-between -5°C and -25°C. The model from Zhang et al. (2010) supports that at $T_{soil}$ colder than -5°C, LWC is negligible.*

[3] An important finding reported is the shift from temperature-dependent CO2 fluxes at sub-zero temperatures to liquid-water dependent fluxes at zero curtain conditions. The authors should here also reflect on the potential bias that this was only observed at one of the 4 sites and consequently this result may be affected also by site-specific differences. Particularly because this is also the warmer and boreal site, where the other sites are all arctic.

In general, the authors focus a lot on RMSE of the different models, but reflect less on the other estimated parameters. E.g. the temperature dependency parameter (B) in figure 5 and 6 differ strongly but a discussion of the potential impact of these differences is lacking. I also suggest relating RMSE, e.g. also in the abstract to mean/median or e.g. seasonal fluxes in order to be able to judge the error in more relative terms.

It is correct that caution should be taken about overgeneralizing our findings from a single site with measurements in zero-curtain conditions. We clarified it in the manuscript. It is also correct that the temperature-dependency of $F_{CO2}$ to $T_{soil}$ changed

significantly between the two regimes (i.e., $T_{soil} < 0^oC$ and zero-curtain conditions). Following this comment, we pointed out that it is clearly displayed by the disparities in the temperature-dependency parameter B. This is the reason why it was not possible to fit a single regression over the measurements at $T_{soil} < 5^oC$ and $T_{soil} \approx 0^oC$. RMSE was added as a percentage of mean $F_{CO2}$.

P11, L337-340: *While the first regime mostly corresponds to Arctic study sites, the second regime only includes one study site (MM) located in the southern boreal forest. Therefore, conclusions from the second regime should be less generalized than those from the first regime.*

P13, L386-393: $F_{CO2}$ increases *more* rapidly with $T_{soil}$ around freezing point than at $T_{soil} < 5^oC$, which is *shown by the higher temperature-dependency parameter (B = 2.82 $^oC^{-1}$) of the MM site* exponential regression (RMSE = 0.286 gC $m^{-2}$ $day^{-1}$) *compared to the exponential regression of Fig. 5 (B = 0.18 $^oC^{-1}$). This discrepancy in temperature-dependency creates a discontinuity between the measurements at $T_{soil} < 5^oC$ and $T_{soil} \approx 0^oC$ that did not allow for a continuous temperature-dependency regression across all the study sites. The lower RMSE of the exponential regression of Fig. 5 (RMSE = 0.024 gC $m^{-2}$ $day^{-1}$; 70.3% of mean $F_{CO2}$) compared to the exponential regression of the MM site (RMSE = 0.286 gC $m^{-2}$ $day^{-1}$; 112.4% of mean $F_{CO2}$) might be due to the impact of soil LWC at the MM site (see Sect. 3.2.3).*

P14, L404-406: The relationship between LWC and $F_{CO2}$ during the non-growing season at MM (RMSE = 0.137 gC $m^{-2}$ $day^{-1}$; *49.1% of mean $F_{CO2}$*) was stronger than between $T_{soil}$ and $F_{CO2}$ (RMSE = 0.286 gC $m^{-2}$ $day^{-1}$; *112.4% of mean $F_{CO2}$*) …

P16, L460-464: *It should be noted that it would be ill-advised to generalize the relationship between soil LWC and $F_{CO2}$ as it is only based on data from one study site, and it cannot be ruled out that this relationship is site-specific depending on soil and vegetation composition. Nevertheless,* our study highlighted the important impact of LWC on $F_{CO2}$ around soil freezing point when there is a mixed state of ice and free water in soils.

P16, L469-471: *Further research on non-growing season $F_{CO2}$ in zero-curtain conditions should investigate different sites to assess if the relationship between $F_{CO2}$ and soil LWC is site-specific or dependent on soil properties.*

[5] In addition, as a suggestion, the authors could consider if a combined model, taking into account both soil T and LWC at the same time could be even better than the presented alternative models only taking one or the other variable into account. After all, both temperature and liquid water are co-limiting the CO2 fluxes but to different extent in the two temperature regimes. If such a combined model could work for both sub-zero and zero curtain conditions it would be a robust model to use for winter conditions in general in the boreal/arctic region.

This is a good suggestion. However, measurements under zero-curtain conditions were only collected at a single study site, a combined model looking at $T_{soil}$ and soil LWC would only represent that one site. To avoid overgeneralizing the results from this site, we did not produce a combined model for both $T_{soil}$ and LWC as it would not take into account potential site-specific differences. To create such a model with confidence, measurements during zero-curtain conditions from other sites would be required. It should be noted that the Random Forest model generated in Sect. 3.2.1 is a model that combined the $T_{soil}$ and LWC effects, although we only exploit it to get an estimate of the predictors' relative importance (i.e., environmental controls) as the authors (and also as proposed by the reviewers) do not want to overgeneralize results from the MM study site.

Specific comments:

[1] L29-30: Exponential relationship with temperature is expected – but they differ in the two situations. Why do you report only RMSE here? Yes, it indicates that it is a better model but why not a line on what was the effect of liquid water availability (i.e. how was the model characterized)?

RMSE was used as a metric to evaluate and compare the regressions as correlation coefficient cannot be used with exponential regressions. More details about the results were added in the abstract.

P1, L29-33: We observed exponential regressions between $CO_2$ fluxes and soil temperature in *fully* frozen soils *(RMSE = 0.024 gC m$^{-2}$ day$^{-1}$; 70.3% of mean $F_{CO2}$)* and soils around freezing point *(RMSE = 0.286 gC m$^{-2}$ day$^{-1}$; 112.4% of mean $F_{CO2}$). $F_{CO2}$ increases more rapidly with $T_{soil}$ around freezing point than at $T_{soil} < 5^oC$. In zero-curtain conditions, the strongest regression was found with soil liquid water content (RMSE = 0.137 gC m$^{-2}$ day$^{-1}$; 49.1% of mean $F_{CO2}$).*

[2] L119: Is a more natural order of the sections here to switch to have data collection before CO2 flux calculations? Consider the same in the result section.

It was chosen to put the theoretical framework of the snowpack diffusion gradient method before discussing data collection to allow the reader to understand why we are measuring certain environmental parameters that might not be obvious otherwise. We understand that there are divergences among researchers about the order of presenting the theoretical framework and data collection, but we think that in the framework of our study, the proposed order is adequate.

[3] L129: NWT not explained

NWT is now spelled out.

P4, L131-133: Trail Valley Creek (TVC), *Northwest Territories*, situated just north of the treeline in the transitional zone between the boreal and Arctic biomes close to the

Mackenzie delta, is dominated by erect-shrub tundra with remaining tree patches (Martin et al., 2022).

[4] L152: In Jones et al 1999, d[CO2]/dz has the unit of ppmv m-1 and you are shifting from ppmv to g C m-3. Also you do not explain that z is in meters and that is how you get to this unit. Please use one line to explain this.

Details were added on how to convert gas concentration units (i.e., ppm to $gC\ m^{-3}$) and how interpret a vertical $CO_2$ diffusion gradient.

P5, L154-157: *Consequently, a vertical $CO_2$ diffusion gradient is maintained through the snowpack (d[$CO_2$]/dz; $gC\ m^{-4}$), with $CO_2$ concentration ([$CO_2$]; $gC\ m^{-3}$) decreasing with snow height from the soil surface (z; m) (Jones et al., 1999). Hereafter, [$CO_2$] is expressed in $gC\ m^{-3}$ but units of concentration could also be expressed in relative units (i.e., ppm) using the ideal gas law.*

[5] L176: D usually should have the unit of m2 time-1 (often seconds but in your case recalculated to daily). You should state the unit, so that the reader can follow how the resulting unit for FCO2 arises.

The units of D were spelled out next to the equation. Note that the unit is also spelled out a few lines earlier.

P6, L177-179: Standard diffusion coefficients of $CO_2$ *(unit: $m^2\ day^{-1}$)* are available in literature but must be corrected for temperature and pressure (Marrero and Mason, 1972; Massman, 1988):

[6] L189: Did you sample from the top of the snow pack first and pushed the sampling rod deeper? Please explain in detail.

Indeed, the gas samples were collected from top to bottom by pushing the sampling rod downward. It was clarified.

P7, L196-199: Gas present in snow pores was collected with a thin, hollow stainless-steel rod (50-120 cm long, 4 mm outer diameter and 2 mm inner diameter) *starting with gas samples in the upper snowpack and then pushing the sampling rod downward to collect gas samples deeper in the snowpack* to minimize snow disturbance (Fig. 2a).

[7] L218: How many different snow densities were measured in the different profiles? And how exactly did you sample snow for density estimation? Please give more info on that.

The snowpack density was measured every 5 cm, it was specified in the manuscript. Although the density of the depth hoar and wind is typically different, the diffusion gradient was still linear. Therefore, the snowpack average density was used for the calculation.

P8, L222-229: Snow properties *were measured at every 5 cm including* snow temperature (Snowmetrics digital thermometer; Fort Collins, Colorado; tenth of a degree resolution), snow density (Snowmetrics digital scale, 100 and 250 $cm^3$ snow cutters *used to weigh snow samples*; $\sigma(\rho snow) \approx 9\%$; Proksch et al., 2016)*, snow liquid water content (hand test from Fierz et al., 2009)* and snow stratigraphy. $T_{soil}$ was measured at 1 cm depth under the soil/snow interface *(Snowmetrics digital thermometer; Fort Collins, Colorado; tenth of a degree resolution)*, three measurements of $T_{soil}$ were averaged. Snow depth measurements were done with a ruler graduated every 1cm ($\sigma(d_{snow}) \approx 0.5$cm).

[8] L243: – snowpit measurements – more precisely snow density measurements, right? Or do you mean all three (density, porosity, tortuosity?) please clarify.

It was clarified that the snowpit measurement uncertainty sources come from snow density and temperature. Afterward, snow density uncertainty propagates to porosity and tortuosity (Eq. 2 and 3) and snow temperature uncertainty propagates to the diffusion coefficient (Eq. 4).

P8, L251-253: Uncertainties can be subdivided into four sources: gas concentration estimates, gas transfer/transport/storage, evaluation of the snowpack $d[CO_2]/dz$ and snowpit measurements *(i.e., snow density and temperature).*

[9] L251: Zero-curtain conditions – it's a little unclear here whether you are defining this term here – which I believe you are. I suggest rephrasing to "Zero-degree Celsius curtain conditions exist when the soil temperature is around freezing point (0°C) for longer periods of time and a mix…" – and maybe even shortly explain in the abstract too as not all readers will be familiar with this term.

It was clarified that the first sentence of Sect. 2.3 is aiming at defining the Zero-curtain conditions.

P1, L25-27: We identified that soil temperature as the main control of non-growing season CO2 fluxes with 68% of relative model importance, except when soil liquid water occurred during zero-degree Celsius curtain conditions (*i.e.,* Tsoil $\approx$ 0°C and liquid water coexist with ice in soil pores).

P9. L263-265: Zero-degree Celsius curtain conditions *exist when* the soil temperature is around freezing point (0°C) and a mix of ice and liquid water coexists in the soil pore space because the phase transition between water and ice is slowed due to latent heat (Outcalt et al., 1990).

[10] L258: It is unclear if you mean that you estimated LWC at all the other sites this way. This is critical to clarify.

LWC was only calculated at MM and was estimated as negligible (i.e., LWC $\approx$ 0) for all other sites. This assumption is supported by the model from Zhang et al. (2010) and served as input for the Random Forest model. This assumption was spelled out in the manuscript.

P9, L267-273: LWC was only monitored at the MM site since it was the only site where $T_{soil}$ *in upper layers remained* around 0°C for the whole *non-growing season*, allowing liquid water in the soil throughout the *non-growing season*. The Zhang et al. (2010) empirical soil liquid water and ice mixing model was used to calculate soil liquid water content ($m_{uw}$) *(Eq. 5 to 8). LWC was estimated to be negligible at the CB, TVC and HPC sites since $T_{soil}$ was in-between -5°C and -25°C. The model from Zhang et al. (2010) supports that at $T_{soil}$ colder than -5°C, LWC is negligible.*

[11] L281: "…based on a **multitude of** decision trees".

Corrected.

P10, L295-296: Random forest (RF) is an ensemble machine learning method based on a *multitude of* decision trees (Breiman, 2001).

[12] L298: Since a major part of the uncertainty is related to density, then it is even more important to know exactly how it was sampled.

See answer to specific comment [7].

[13] L305: in principle your figure legend symbol indications are not entirely correct, but I see the point. Maybe add to figure text that colors indicate site and symbol type indicate year…

Clarifications about the legend were added to the figure text.

P11, Fig. 3: *The data dots color indicates the study site and its symbol (i.e., circle or x-shaped) indicates the non-growing season during which it was collected.*

[14] L320: But these regimes coincide with regime 1 being the northern sites and regime 2 being MM – seems like a major confounding factor of this cross site analysis, as site-specific conditions may

See answer to general comment [2].

[15] L328: SWE only explained first time a few lines later…

The definition of SWE was moved to its first mention.

P11-12, L344-349: $T_{soil}$ was the $F_{CO2}$ predictor with the highest relative importance (68%) when using the complete dataset (Fig. 4a), followed by LWC (17%). Snowpack characteristics, $\rho_{snow}$ (11%) and *snow water equivalent (SWE)* (2%), had a lower relative importance in the RF model. Contrary to what might be expected, the vegetation type had near-negligible relative importance (1%) in $F_{CO2}$ prediction. The RF model was developed starting with all environmental variables available: $T_{soil}$, LWC, vegetation

type, *SWE*, snow depth, mean $\rho_{snow}$, max $\rho_{snow}$, $\varphi$, $\tau$, wind slab fraction and wind slab thickness.

[16] Also – how can you include LWC in a full model across all sites when it was only measured at the MM site? Connected to my question above if you estimated LWC at the other sites with the method described above.

See answer to specific comment [10].

[17] L410: Figure 8 – shrubs, not schrubs.

It was corrected in Fig. 8.

[18] L421: confirms not corroborates

Corrected.

P16, L444-446: Our results *confirm* the strong non-growing season $F_{CO2}$ dependency on $T_{soil}$ shown by Natali et al. (2019), although we observed fluxes lower than reported by Natali et al. (2019) at $T_{soil} < -5°C$ and mostly higher fluxes at $T_{soil} > -5°C$.

[19] L445: soil pores not soil porosities

Corrected.

P16-17, L474-477: It is well known that anaerobic conditions created by high soil moisture (at least $> 50\%$) constrain soil $CO_2$ respiration rates during the growing season because many microorganisms require oxygen for organic matter decomposition which they lack if soil *pores* are filled with water (Linn and Doran, 1984; Davidson and Janssens, 2006).

[20] L463: availability and **quality** of labile C

Added.

P17, L495-498: $CO_2$ production is governed by the availability *and quality* of labile C compounds regulating the decomposition of soil organic matter (Michaelson et al., 2005; Wang et al., 2011), and the activity and composition of the soil microbial community (Monson et al., 2006).

[22] L990: Figure A1 + A3: should be 'shrubs' – not 'schrubs'

It was corrected in Fig. A1 and A3.

---

## Author Comment (AC3)

In blue: Reviewer's comments. [ ] = Numbering

In black: Answers to referees.
P=Page; L=Line; Track change version
*In black and italic: Modification added to text.*

**Reviewer #1:**

Synopsis:

The manuscript by Mavrovic et al. is a nice investigation on the controls of CO2 emissions from high latitude soils during the cold season. Although the study is not showing many things that are particularly new, there are a few insights here that I like. And since there's little data from the cold season in these environments, the data presented in this study is going to quite valuable to a wider audience. Therefore, I think it can be suitable for publication after some adjustments and clarifications.

Thank you for this assessment of our study.

First of all, one of the main conclusions is that non-growing season fluxes during the zero curtain are controlled by liquid water content, and not by temperature. That's not particularly surprising since soil temperatures do not vary during the zero curtain period, so other variables should be more important. Likewise, the authors say that liquid water content is less important than soil temperature when liquid water content is low, but that's pretty much the same thing: it's a subset of the data where there's little change in liquid water content. The authors acknowledge these aspects briefly in their results, but it should be discussed more extensively in the discussion. What is the relative importance of liquid water content? Figure 7 suggests that it's responsible for a huge change in the fluxes.

Based on this comment, Sect. 4.1 (Controls of non-growing season $CO_2$ fluxes) has been fleshed out to discuss further the respective impact of soil temperature and liquid water content on $CO_2$ fluxes. The Random Forest model generated in Sect. 3.2.1 (Fig. 4) is used to estimate the predictors' relative importance (i.e., environmental controls).

P16, L441-448: The RF model *predictors' relative importance showed* that *during the non-growing season, $T_{soil}$ emerged as the dominant predictor of $F_{CO2}$ when the soil was frozen. Nevertheless, in the closed-boreal forest site (i.e., MM) where zero-curtain conditions persisted throughout the non-growing season, soil LWC took precedence as the dominant predictor as there was minimal variation in $T_{soil}$ under these conditions.* Our results corroborate the strong non-growing season $F_{CO2}$ dependency on $T_{soil}$ shown by Natali et al. (2019), although we observed fluxes lower than reported by Natali et al. (2019) at $T_{soil} < -5^oC$ and mostly higher fluxes at $T_{soil} > -5^oC$. *Considering the two regressions of the relationship between $T_{soil}$ and $F_{CO2}$ have large uncertainties attached to them, the difference between them falls inside the uncertainty margin (Fig. 5).*

General comments:

[1] Otherwise, it's strange to see so few parameters being part of the random forest analysis. Why weren't more soil properties, like C:N ratios part of this analysis? Or calculated available pore space? There's a huge variation in the fluxes, and there are few soil or site-specific differences that are addressed in this manuscript. Other causes for the peaks seen at below-zero temperatures are not discussed, even though it has been discussed in the past that soil-freezeup reduces available pore space, which leads to bursts of greenhouse gases in permafrost environments (see e.g. Mastepanov et al 2013).

All soil variables available were included in the analysis (i.e., suface soil temperature and soil liquid water content). In this study, we aimed to get several study sites and sampling locations to obtain a good overview of the spatial variability. This contrasted with most other studies, which focused on a limited number of study sites (Pork et al., 2016; Webb et al., 2022). Considering the workload associated with winter data collection during harsh field conditions at the selected sites, we had to limit our data collection to the presented data of $CO_2$ soil gases, snow and environmental measurements. Further, since we conducted our $CO_2$ flux measurements during winter at most study sites, the soil was too frozen to collect soil samples for laboratory analysis. Consequently, it is acknowledged in the discussion that soil properties and biogeochemistry was not included in our analysis and might explain the unaccounted variability in our results.

In regards to peaks seen at below-zero temperatures, see answer to specific comment [10].

P17, L493-495: The unexplained variance (16%) suggests that non-growing season $CO_2$ fluxes might have been controlled by other environmental variables such as soil physical-chemical properties regulating soil biogeochemistry and soil redox conditions, which were not addressed *nor measured* in this study.

[2] Finally, the authors compare their results to those from Natali et al. (2019), and then discuss how their regression is different below and above -5 degrees C. However, both regressions have large uncertainties attached to them, so this difference is most likely non-significant. If you think your result is different, please show that with a statistical test.

It is true that the difference between our exponential regression and the one from Natali et al. (2019) falls inside the uncertainties of both regressions. This point was spelled out more clearly in the manuscript.

P16, L447-448: *Considering the two regressions have large uncertainties attached to them, the difference between them falls inside the uncertainty margin (Fig. 5).*

Specific comments:

[1] The title does not show that these are only soil fluxes. Please add that detail. Also, I would say 'cold season' or 'winter' rather than 'non-growing season' since you clearly measured in the middle of winter.

The title was modified to specify that we are looking at soil carbon fluxes. The term non-growing season was chosen instead of winter as it is commonly used in carbon science and more clearly defined. By clarifying that this study focused on soil carbon fluxes, it also implies that we remove the influence of photosynthesis to focus on soil respiration. The terminology was uniformized to non-growing season throughout the manuscript.

Modified title: Environmental controls of non-growing season *soil* carbon dioxide fluxes in boreal and tundra environments

P3-4, L113-115: Spatio-temporal measurements of snowpack $CO_2$ diffusion gradients were performed at several locations in four sites during the 2020-2021 and 2021-2022 *non-growing seasons* (December to May).

P7, L192-193: All data were collected during the 2020-21 and 2021-22 *non-growing seasons* between December and May (Table 1).

P8, L238-239: Randomly distributed gas samples collected during the 2020-21 *non-growing season* were analyzed with a Picarro G2201-I CRDS gas analyzer (Picarro, Santa Clara, Californie; $\sigma < 0.1\%$; N = 26).

P9, L267-269: LWC was only monitored at the MM site since it was the only site where $T_{soil}$ *remained around 0$^o$C for the whole non-growing season,* allowing liquid water in the soil throughout the *non-growing season*.

P11, Fig. 3: $CO_2$ flux ($F_{CO2}$) uncertainty relationship to $F_{CO2}$ for the four study sites and two *non-growing seasons* 2020-2021 and 2021-2022. Specifications of the linear fit can be found in the upper left. The data dots color indicates the study site and its symbol (i.e., circle or x-shaped) indicates the *non-growing season* during which it was collected.

P14, Fig. 6: $CO_2$ flux ($F_{CO2}$) as a function of soil temperature ($T_{soil}$) at the Montmorency Forest study sites where soil liquid water content (LWC) was greater than 0 m$^3$/m$^3$ through the *non-growing season*. An exponential regression was fitted to the data (black line).

P15, L420-421: Higher $F_{CO2}$ can be explained by warmer mean annual average temperature, a deeper snowpack and *non-growing season* $T_{soil}$ around 0$^o$C (See Sect. 3.4).

P16, L441-444: The RF model supported that *during the non-growing season, $T_{soil}$ emerged as the dominant predictor of $F_{CO2}$ when the soil was frozen. Nevertheless, in the closed-boreal forest site where zero-curtain conditions persisted throughout the non-*

*growing season, soil LWC took precedence as the dominant predictor as there was minimal variation in T$_{soil}$ under these conditions.*

P16, L458-459: Soil LWC was observed only at the MM site, where T$_{soil}$ was around 0$^o$C throughout *the non-growing season*.

P17, L479-484: Our study shows that abiotic variables related to T$_{soil}$, LWC, and physical snowpack properties explain the majority of variance in *non-growing season* CO$_2$ fluxes. It should be noted that we did not incorporate variables related to temporal dynamics such as the previous days' soil temperature and LWC, which have been shown by Harel et al. (2023) to be of importance during the growing season. However, *non-growing season* soil variables are not expected to be as dynamic as during the growing season because of the snowpack insulating properties.

P17, L493-495: The unexplained variance (16%) suggests that *non-growing season* CO$_2$ fluxes might have been controlled by other environmental variables such as soil physical-chemical properties regulating soil biogeochemistry and soil redox conditions, which were not addressed in this study.

P18, L523-524: However, we found that at our site maintaining zero-curtain conditions throughout *the non-growing season*, LWC becomes the main control of non-growing season F$_{CO2}$.

[2] Page 6, Equation 2: it's a minor detail but this is only true if there's no liquid water in the snowpack. Looks like your sites did not experience melt events, so this is probably not an issue. Could be mentioned though.

There are some measurements at the Montmorency Forest site where there was liquid water in the snowpack. The more general equation for snow porosity was used in this instance. It was corrected in the manuscript.

P6, Eq. 2: $\varphi = 1 - \frac{\rho_{snow}}{\rho_{ice}} + \theta \cdot \left( \frac{\rho_{snow}}{\rho_{ice}} - 1 \right)$

P6, L169-170: where $\rho$ represents the density of snow and pure ice ($\rho_{ice} = -0.0001 \cdot T_{ice} + 0.9168$ with T$_{ice}$ as ice temperature in $^o$C and $\rho_{ice}$ in g cm$^{-3}$; Harvey et al., 2017) *and $\theta$ is the snow liquid water content*.

P8, L222-226: Snow properties *were measured at every 5 cm* including snow temperature (Snowmetrics digital thermometer; Fort Collins, Colorado; tenth of a degree resolution), snow density (Snowmetrics digital scale, 100 and 250 cm$^3$ snow cutters *used to weigh snow samples*; $\sigma(\rho_{snow}) \approx 9\%$; Proksch et al., 2016)*, snow liquid water content (hand test from Fierz et al., 2009)* and snow stratigraphy.

P22, L691-693: *Fierz, C., R. L., A., Durand, Y., Etchevers, P., Green, E., McClung, D., Nishimura, K., Satyawali, P., and Sokratov, S.: The International Classification for*

*Seasonal Snow on the Ground, IHP-VII Technical Documents in Hydrology N83, IACS Contribution N1, UNESCO-IHP, Paris, 2009.*

[3] Line 217-218: did you determine average snow density for the snowpack or did you make a profile? I wonder how the density differences between the depth hoar and wind slab affects your calculations.

The snowpack density was measured every 5 cm, as we specified in the manuscript. Although the density of the depth hoar and wind is typically different, the diffusion gradient was still linear. Therefore, the snowpack average density was used for the calculation.

P8, L222-229: Snow properties *were measured at every 5 cm* including snow temperature (Snowmetrics digital thermometer; Fort Collins, Colorado; tenth of a degree resolution), snow density (Snowmetrics digital scale, 100 and 250 $cm^3$ snow cutters *used to weigh snow samples*; $\sigma(\rho snow) \approx 9\%$; Proksch et al., 2016)*, snow liquid water content (hand test from Fierz et al., 2009)* and snow stratigraphy. $T_{soil}$ was measured at 1 cm depth under the soil/snow interface *(Snowmetrics digital thermometer; Fort Collins, Colorado; tenth of a degree resolution)*, three measurements of $T_{soil}$ were averaged. Snow depth measurements were done with a ruler graduated every 1cm ($\sigma(d_{snow}) \approx 0.5cm$).

[4] Line 219-221: why only measure Tsoil at such a shallow depth? I would expect respiration to be relatively high across the root zone, and temperatures may differ with depth, affecting your correlations.

Soil temperature was measured at a shallow depth because it was not possible to go deeper in frozen soil and no permanent sensors were installed at the individual sampling locations. It is now mentioned un the manuscript.

P8, L226-229: $T_{soil}$ was measured at 1 cm depth under the soil/snow interface *as it was not possible to go deeper in frozen soil and no permanent sensors were installed* (Snowmetrics digital thermometer; Fort Collins, Colorado; tenth of a degree resolution), three measurements of $T_{soil}$ were averaged.

P16, L454-456: *It should be reminded that the $T_{soil}$ used in his study refer to near-surface temperature, deeper $T_{soil}$ may vary and affect the correlation with $F_{CO2}$.*

[5] Line 223: why only 86%?

The rest of the gas samples were processed by an independent lab (Groupe de recherche interuniversitaire en limnologie, Université de Montréal) to validate the method that was used at the Université du Québec à Trois-Rivières lab for most of the samples. It was clarified in the manuscript.

P8, L238-242: Randomly distributed gas samples collected during the 2020-21 non-growing season were analyzed with a Picarro G2201-*i* CRDS gas analyzer (Picarro, Santa

Clara, Californie; σ < 0.1%; N = 26) *to validate the method used with the LI-7810 to determine $CO_2$ concentration.* $CO_2$ concentrations estimated from the LI-7810 and Picarro G2201-*i* gas analyzers were not significantly different in their concentration range and distribution (Fig. A2; $R^2 = 0.92$).

[6] Line 241: these are only random errors related to your calculations, assuming that the method is perfect. Which systematic errors may have affected your measurements?

It was clarified that Sect. 2.2.3 focuses on random errors while systematic errors are discussed at the end of Sect. 2.2.1.

P6, L183-190: The diffusion gradient method assumes that gas fluxes are the result of simple, linear, gradient-induced diffusion in uniform porosity through snow cover (McDowell et al., 2000). A snowpack with strongly heterogeneous density (i.e., vertical stratification) can induce a bias when gas flow is altered by dense layers or ice crusts, typically leading to $F_{CO_2}$ overestimation (Seok et al., 2009). Such layers were rarely found in our study sites. The diffusion gradient assumption also does not hold when strong wind events occur, decreasing snowpack $CO_2$ concentration through wind-pumping and inducing a negative bias on CO2 fluxes (Seok et al., 2009). Consequently, $d[CO_2]/dz$ was not measured in days following a strong wind event.

P8, L249-250: *The uncertainty assessment focuses on random errors, as systematic errors are discussed at the end of Sect. 2.2.1.*

[7] Line 248-249: the uncertainty analysis is not explained well. What is the min-max uncertainty propogation method? Please elaborate, and give a reference.

Details were added on the uncertainty analysis as follow:

P9, L258-261: *$F_{CO_2}$ uncertainty was estimated by propagation of the uncertainties of $d[CO_2]/dz$ and snow density using Eq. 1 (Taylor, 1997). The uncertainty of ρsnow was fixed at 9% (Proksch et al., 2016) while the uncertainty of $d[CO_2]/dz$ was estimated based on the root mean squared error of the linear regression for each snowpack concentration gradient measurement.*

P29, L960-961: *Taylor, J.R.: An Introduction to Error Analysis: The Study of Uncertainties in Physical Measurements. 2nd Edition, University Science Books, Sausalito, United States, 343 pages, ISBN-10: 093570275X, 1997.*

[8] Line 256-258: this is written a bit strangely. Zero curtain happens at all your sites, you simply were not there to measure it.

The point we wish to convey is that Montmorency Forest is the only site where zero-curtain conditions last the whole non-growing season, while it only occurs during shoulder season freezing and thawing at the other sites. The sentence was rephrased for clarity.

P9, L267-270: LWC was only monitored at the MM site since it was the only site where $T_{soil}$ *remained around 0°C for the whole non-growing season,* allowing *the presence of* liquid water in the soil throughout the *non-growing season*.

[9] Line 354: which lack of measurements between -0.5 and -6? There a few. Or do you mean the number is too low?

We meant that the number of measurements is low, it was clarified in the manuscript as follows:

P12, L371-372: Note that the *low number* of $F_{CO2}$ measurements with $T_{soil}$ between -6°C to -0.5°C restrict the capacity to evaluate the regression within this range.

[10] Line 362-363: like I mentioned before, this may be due to changes in available porosity. See also Pirk et al. (2015) for how this works with methane fluxes (but same principal holds for CO2). Other useful papers are Zona et al. (2016) and Raz-Yaseef et al. (2017), who showed similar bursts with eddy covariance. Otherwise, I wonder whether changes in air pressure may have played a role, which may affect the storage in deep snow packs.

Unlike Pirk at al. (2015), who observed $CH_4$ bursts during autumn freeze-in, we observed a few high $F_{CO2}$ during winter when $T_{soil}$ was between -25°C and -10°C. It seems unlikely that much ice formation occurs at those $T_{soil}$, but maybe some soil cracking. This hypothesis was added to the manuscript.

The challenge of air pressure change is addressed by Seok et al. (2009). Change in air pressure is strongly correlated with the wind speed. Changes in air pressure and wind speed decrease snowpack $CO_2$ concentration inducing a negative bias on $CO_2$ fluxes. There are still uncertainties on the impact of changes in air pressure and wind on the gas diffusion method, Seok et al. (2009) proposed corrections that can be applied to account for strong winds. In our study, we did not conduct measurements close to strong wind events to account for this uncertainty. No disparities were observed in the range and average of our measurements over time during the course of our two-week campaigns, which indicates that no drastic changes in snowpack $CO_2$ concentration seem to have occurred during our measurements.

P13, L380-384 *It has been suggested that gas bursts during autumn freeze-up in permafrost environments might be due to gas compression by ice formation and ground cracking (Pirk et al. 2015). This hypothesis can be considered to explain the high FCO2 observed in this study, although the high $F_{CO2}$ observed occurred at a near-surface $T_{soil}$ between -25°C and -10°C so the freeze-up would have to occur at lower depths in the soil.*

P26-27, L876-879: *Pirk, N., Santos, T., Gustafson, C., Johansson, A., Tufvesson, F., Tamstorf, Parmentier, F.-J., Mastepanov, M., and Christensen, T.: Methane emission*

*bursts from permafrost environments during autumn freeze-in: New insights from ground-penetrating radar. Geophysical Research Letters, 42(16), 6732-6738, doi: 10.1002/2015GL065034, 2015.*

[11] Line 425-427: but Natali et al. also had many more datapoints, so their estimate is better constrained. Anyway, I doubt there is a statistically significant difference between the regressions in your two studies.

It is true that the difference between our exponential regression and the one from Natali et al. (2019) falls inside the uncertainties of both regressions. This point was spelled out more clearly in the manuscript.

P16, L447-448: *Considering the two regressions have large uncertainties attached to them, the difference between them falls inside the uncertainty margin (Fig. 5).*

---

## Author Response (AR1)

In blue: Reviewer's comments. [ ] = Numbering

In black: Answers to referees.
P=Page; L=Line; Track change version
*In black and italic: Modification added to text.*

**Editor:**

Dear Authors

thank you for carefully answering the comments of the two referees.

I have a few comments:

[1] Both reviewers asked to clarify the calculation of the diffusion coefficient (D) across the snow pack and you argued for a vertically constant D as the measured vertical concentration gradient was linear. On the other hand you find vertical differences in the snow density and the formation of a depth hoar layer. You explain that you have collected snow density every 5 cm to calculate snow porosity, tortuosity and the CO2 diffusion coefficient. I would like you to show the data on the vertical stratification of the snow pack physical state (at least some examples as supplementary material) and the local D values and see whether or not there is a contradiction between the conclusion from the CO2 concentration profiles (D = const.) and the D calculated for vertical strata from snow pack physical states.

The diffusion coefficient depends on temperature and only changes by a few percent across temperatures between 0ºC and -20ºC (Eq. 4, P6). Snow porosity and tortuosity depend on snow density, which displays vertical stratification. An average snow density, and therefore an average porosity and tortuosity, was used in $CO_2$ flux calculations (Eq. 1, P6) since the diffusion gradient remained linear despite vertical stratification in snow density. Figure A3 was added in Appendix A to display a few examples of typical snowpack concentration gradients at each study site along with snow density stratification.

P7, L224-225: *Examples of snow density vertical stratification along with $CO_2$ concentration measurements can be found in Appendix A (Fig. A2).*

P31, Figure A2: *Examples of snow density (ρsnow) vertical stratification and $CO_2$ concentration ([$CO_2$]) gradient measurements in function of snow height ($h_{snow}$) from the ground level. The coefficient of determination ($R^2$), [$CO_2$] gradient (m) and y-axis intercept (b) for the linear regressions on the [$CO_2$] gradient measurements are provided. Data from (a) Montmorency Forest balsam fir closed-crown coniferous boreal forest on 2021-02-26, (b) Cambridge Bay prostrate-shrub tundra (hydric tundra: hydric sedge fen) on 2022-04-15, (c) Trail Valley Creek erect-shrub tundra (lichen) on 2022-03-26, and (d) Havikpak Creek black spruce open-crown coniferous boreal forest on 2022-03-16.*

[2] The comment of Referee 1 on the title and especially the examined period is very relevant and I strongly recommend you do reconsider your decision.
- You do neither define the growing season nor the non-growing season
- Even if you did, the non-growing season might span wider than what you observed
- Your discontinuous measurements do not cover a whole season anyway, but are rather examples for situations with snow.
- Your work is very much on snow , isn't it? So why not using one of the following: "cold season", "winter" or even better "snow period" in the title and the abstract ?

The examined period was replaced from "non-growing season" to "winter" throughout the manuscript. This is coherent with published articles such as Natali et al. (2019), Björkman et al. (2010), Kim et al. (2019), Monson et al. (2006), Sturm et al. (2005) and Wang et al. (2011).

Title: Environmental controls of *winter* soil carbon dioxide fluxes in boreal and tundra environments

[3] Your answer on atmospheric pressure (p_a) changes raised by Referee 1 is incomplete: The pressure does not only change with wind but at synoptic time scales with high and low pressure systems passing the site, see, e.g., Kissas et al. (2022) (for illustration, no need to cite this paper if you do not deem it relevant). Please refer to meteorological data products (e.g. ERA5) to examine whether such event had resolved the observed peak emissions. in that case the mechanisms might even be comparable with Kissas' study.

A few sampling locations at two study sites (Trail Valley Creek and Cambridge Bay) were revisited daily or every few days over a few weeks. No major disparities were observed in the range of $F_{CO2}$ measurements over time at those sampling locations despite fluctuations in wind speed and atmospheric pressure. Figure A1 was added as an example of such measurements in Appendix A. We also added text in the manuscript.

P6, L186-188: *Monitoring of $F_{CO2}$ at a few sampling locations did not show any relationship between $F_{CO2}$ and wind speed or atmospheric pressure (e.g., Fig. A1).*

P30, Figure A1: *$CO_2$ fluxes ($F_{CO2}$) at a sampling location in the Trail Valley Creek erect-shrub tundra (lichen) between March 19th and March 27th, 2022. Atmospheric pressure and wind speed were obtained from Environment and Climate Change Canada's Meteorological Service of Canada meteorological station at Trail Valley Creek (https://climate.weather.gc.ca/historical_data/search_historic_data_e.html).*

[4] in your revised section "P17, L493-495:" consider changing "were *not* addressed nor measured in this study" to " were *neither* addressed nor measured in this study".

Modified.

P17, L489-491: The unexplained variance (16%) suggests that non-growing season $CO_2$ fluxes might have been controlled by other environmental variables such as soil physicalchemical properties regulating soil biogeochemistry and soil redox csonditions, which were *neither* addressed nor measured in this study.

---

## Author Response (AR2)

In blue: Reviewer's comments. [ ] = Numbering

In black: Answers to referees.
P=Page; L=Line; Track change version
*In black and italic: Modification added to text.*

**Editor:**

Dear Authors
thank you for your clarification and corrections of the revised manuscript.

I'd like to come back to my first question, which was not fully answered.

I asked for
"I would like you to show the data on the vertical stratification of the snowpack physical state (at least some examples as supplementary material) and the local D values and see whether or not there is a contradiction between the conclusion from the $CO_2$ concentration profiles (D = const.) and the D calculated for vertical strata from snow pack physical states."

You did not yet provide " local D values" and did not explain "whether or not there is a contradiction between the conclusion from the $CO_2$ concentration profiles (D = const.) and the D calculated for vertical strata from snow pack physical states."

Your answer on the local D related solely to the small effect from snow temperature but not on effects of snow and soil properties such as snow porosity and tortuosity. (see lines 226 and 227: "Once the gas samples were collected, a vertical profile of snow and soil properties was measured to calculate snow porosity, tortuosity and the $CO_2$ diffusion coefficient") .

This sentence can be interpreted in a way that you calculated local D values from snow porosity and tortuosity (see, e.g. Albert, M. R. and E. F. Shultz (2002). "Snow and firn properties and air–snow transport processes at Summit, Greenland." Atmospheric Environment 36(15): 2789-2797., where the diffusion coefficient depended on porosity, their eq(1), and not only on temperature.

I do not ask for an extension of your work, but for clarification and reformulation, if you didn't calculate $CO_2$ diffusion coefficients including the effects from porosity and tortuosity . In your discussion I'd like to ask you to comment on the apparent contradiction, that the D can be found constant in a profile with large vertical variation og porosity and tortuosity values.

I hope this is now clearer, otherwise please ask.

We propose a better idea to clarify the misunderstanding about the diffusion coefficient. The diffusion coefficient in our manuscript is the air diffusion coefficient ($D_a$ in Albert and

Shultz 2002) which only depends on air temperature. In Albert and Shultz (2002), a snow diffusion coefficient ($D_s$) is defined to include the effect of porosity and tortuosity on $D_a$. We did not use a $D_s$ since we included the explicit effect of porosity and tortuosity in our main $CO_2$ flux equation (Eq. 1). We see how it was indeed confusing. We hence clarified in the manuscript that we are referring to $D_a$ and not $D_s$. Fig. A3 in Appendix A was modified to display $D_s$ stratification instead of snow density stratification. Fig. A3 is used to support some comments on the $D_s$ stratification that were added in the discussion (Sect. 4.3 Snowpack Importance).

P5, Eq. 1: $F_{CO2} = -\varphi\tau D_a \frac{d[CO_2]}{dz}$

P5, L159-160: where $\varphi$ represents the porosity of the snow medium, $\tau$ its tortuosity and $D_a$ the *air* diffusion coefficient of the diffused gas in $m^2$ $day^{-1}$.

P6, L173-175: Standard *air* diffusion coefficients of $CO_2$ (unit: $m^2$ $day^{-1}$) are available in literature but must be corrected for temperature and pressure (Marrero and Mason, 1972; Massman, 1988):

P6, Eq. 4: $D_a = 0.2020 \cdot \left(\frac{T}{T_o}\right)^{1.590} \cdot e^{-\frac{0.3738}{T/T_o}}$

P7, L219-220: Once the gas samples were collected, a vertical profile of snow and soil properties was measured to calculate the $CO_2$ *air* diffusion coefficient *from the snow temperature*, and snow porosity *and* tortuosity *from snow density*.

P7, L224-225: Examples of snow vertical stratification along with $CO_2$ concentration measurements can be found in Appendix A (Fig. A3).

P17, 487-492: *Regarding the snowpack diffusion gradient method, the snowpack is used to estimate winter $CO_2$ fluxes. An average snow density was used to estimate snow porosity and tortuosity used in $CO_2$ flux calculations (Eq. 1), which does not consider the vertical stratification of the snowpack. However, the diffusion gradient remained linear despite vertical stratification in snow density (e.g., Fig. A3 where the average ratio between the standard deviation and mean of $D_{air} \cdot \varphi \cdot \tau$ is around 10%) which points toward a minimal impact of this assumption on our results.*

P32, Fig. A3: Examples of *snow diffusion coefficient ($D_{snow} = D_{air} \cdot \varphi \cdot \tau$)* vertical stratification and $CO_2$ concentration ($[CO_2]$) gradient measurements in function of snow height ($h_{snow}$) from the ground level. The coefficient of determination ($R^2$), $[CO_2]$ gradient (m) and y-axis intercept (b) for the linear regressions on the $[CO_2]$ gradient measurements are provided. *The ratio between $D_{snow}$ standard deviation ($\sigma(D_s)$) and average ($\overline{D_s}$) is provided in percent.* The data comes from (a) Montmorency Forest balsam fir closed-crown coniferous boreal forest on 2021-02-26, (b) Cambridge Bay prostrate-shrub tundra (hydric tundra: hydric sedge fen) on 2022-04-15, (c) Trail Valley Creek erect-shrub tundra (lichen) on 2022-03-26, and (d) Havikpak Creek black spruce open-crown coniferous boreal forest on 2022-03-16.